



# Changes of Nonlinearity and Stability of Streamflow Recession Characteristics under Climate Warming in a Large Glaciated Basin of the Tibetan Plateau

Jiarong Wang[1, 2], Xi Chen[1, 2]*, Man Gao[1], Qi Hu[3], and Jintao Liu[2]

[1] Institute of Surface-Earth System Science, School of Earth System Science, Tianjin University, Tianjin 300072, P.R. China
[2] College of Hydrology and Water Resources, Hohai University, Nanjing 210098, P.R. China
[3] School of Natural Resources and Department of Earth and Atmospheric Sciences, University of Nebraska-Lincoln, Lincoln NE 68583 U.S.A.

*Correspondence to: Xi Chen, e-mail: xichen@hhu.edu.cn





**Abstract.** The accelerated climate warming in the Tibetan Plateau after 1997 has strong consequences in hydrology, geography, and social wellbeing. In hydrology, the change of streamflow as a result of changes of dynamic water storage originating from glacier melt and permafrost thawing in the warming climate directly affects the available water resources for societies of the most populated nations in the world. In this study, annual streamflow recession characteristics are analyzed using daily climate and hydrological data during 1980–2015 in the Yarlung-Zangpo River basin (YRB) of south Tibetan

Plateau. The recession characteristics are examined in terms of $dQ/dt = -aQ^b$ and the response/sensitivity of streamflow to changes of groundwater storage. Major results show that climate warming significantly increased the nonlinearity of the response ($b$) and decreased streamflow stability [$\log(a)$] in most sub-basins of YRB. These changes of recession characteristics are attributed to opposite effects of increases of available water storage and recession timescale on the recession. Climate warming increased sub-basin water storage considerably by more recharge from accelerated glacier melting and permafrost

thawing after 1997. Meanwhile, the enlarged storage lengthens recession timescales and thereby decreases the sensitivity of discharge to storage. In the recession period when the recharge diminished, increased evaporation under warmer temperatures acts as a competing process to reduce water storage and streamflow. While reservoir regulations in some basins helped reduce and even reverse some of these climate warming effects, this short-term remedy could only function before the solid water storage is exhausted when the climate warming continues.

**Keywords.** Climate warming; Streamflow Recession Characteristics; Change of storage-discharge relationship; Permafrost degradation; Glacier melting



# 1. Introduction

The warming rates of air temperature in high latitudes and high altitudes are greater than the rate of change of global average near surface air temperature (e.g., McBean et al., 2005; Pepin et al., 2015). The greater climate warming has accelerated glacier

melting and permafrost thawing in cold alpine regions, causing significant glacier and permafrost retreats (e.g., Yao et al., 2004, 2007) and landscape alternations (e.g., Niu et al., 2019). These changes could alter hydrodynamics of streamflow and groundwater storage in the alpine regions and their downstream tributaries (Bense et al., 2012; Walvoord and Striegl, 2007; Walvoord and Kurylyk, 2016; Li et al., 2018; Wang et al., 2018; Yi et al., 2021). A recent study of Wang et al. (2021) has shown that such changes have also caused variation of precipitation-streamflow relationship. The societal impacts of these

changes are profound because they redefine freshwater availability and its seasonality for populations of billions in the downstream tributaries (e.g., Cuo et al., 2014; Zhang et al., 2013; Wang et al., 2020).

Many studies have found that compound effects of glacier and permafrost retreats in the past decades had reshaped the groundwater flow and hydrological cycle (e.g., Bring et al., 2016; Forster et al., 2014; Ji et al., 2020; Walvoord and Kurylyk, 2016). The accelerated glacier melting and permafrost thawing have increased the soil active layer thickness (ALT) and

therefore enlarged groundwater storage by allowing exchange of the surface water and groundwater (Xu et al., 2017; Forster et al., 2014; Ji et al., 2020). Such exchange further alters streamflow composition in arctic catchments (Chang et al 2008; Walvoord and Kurylyk, 2016; Bring et al., 2016) and the northeastern and southern Tibetan Plateau (TP) (Li et al., 2018; Wang et al., 2018; Yi et al., 2021).

In those catchments, changes of groundwater storage and subsurface moisture profile due to permafrost retreat could reroute

the subsurface flow paths during low flows (Koch et al., 2014; Payn et al., 2012). The thickened ALT allows infiltration through the previously permafrost layer into aquifers and thus increases recharge in permafrost basins. It has been reported that the permafrost loss in the past decades has enhanced regional groundwater circulation (shortening its timescale) with more discharge to stream flows (Ji et al., 2020; Walvoord et al., 2012). As examples, the baseflow in the source region of the Yangtze River increased at a rate of 1.35 mm·a⁻¹ during 1962–2012 following the annual temperature rise of ~1.32 °C (Yi et al., 2021),

and 1.09 mm·a⁻¹ during 1979–2013 in another glacierized basin of YBJ (Fig. 1), also in the Yangtze River with its annual temperature rise of ~0.98 °C (Lin et al., 2020). Walvoord and Striegl (2007) found that the groundwater contribution to streamflow in an arctic basin increased by 0.7–0.9% per year from the 1950s to 2005 following its annual temperature rise of ~1.24 °C during that period.

Effects of these changes of water budget and subsurface moisture profile on hydrograph are complicated in frozen areas. The

increase of ALT could reduce the buffering effect of soils on streamflow variability and thereby increase the baseflow recession rate (Lyon et al., 2009; Lyon and Destouni, 2010; Brutsaert and Hiyama, 2012). On the other hand, the increase of ALT enlarges groundwater storage which can strengthen aquifer regulations on groundwater flow and slow down the recession rate (Lin et





al., 2020; Mao and Wang, 2016). These effects on the recession rate can result in strong nonlinear behavior of streamflow in time and space.

During periods with little or no precipitation, the baseflow recession (or relationship of d$Q$/d$t$ vs. $Q$, where $Q$ is discharge) is typically quantified by a power law differential equation (Brutsaert and Nieber, 1977; Tallaksen, 1995), i.e., d$Q$/d$t$ = -$aQ^b$. The depletion of baseflow in relation to the parameters $a$ and $b$ contains valuable information concerning storage properties and aquifer characteristics of basins (Tallaksen, 1995). The recession scale parameter $a$ is a function of the hydraulic and geometric properties of the aquifer in a basin and can be used as a proxy for effective depth to permafrost in frozen areas (Lyon and
Destouni, 2010). The parameter $b$ as reflected in the concavity of the hydrograph or the nonlinearity of recession (Dralle et al., 2017) is a function of boundary conditions to describe the equivalent water depth profile of an aquifer (Brutsaert and Nieber, 1977; Tashie et al., 2020). So, $b$ can be interpreted as a measure of the diversity of water transport timescales throughout various parts of a catchment (Harman et al., 2009). Therefore, the variations of $a$ and $b$ in time and space can describe recession characteristics distribution of a basin (e.g., Brutsaert and Nieber, 1977; Kirchner, 2009).

Changes of the recession characteristics in time reflect their vulnerability to climatic and anthropogenic factors (Berghuijs et al., 2016; Brooks et al., 2015; Buttle, 2018). Streamflow stability, log($a$), has a significant seasonal cycle for over 99% of basins examined in the Continental United States (Tashie et al., 2020). Moistening climate in catchments could increase the diversity of flow paths and the nonlinear relationships between storage/recharge and discharge (Brutsaert and Nieber, 1977; Hinzman et al., 2020). In cold climate regions, reduced glacier size can lead to considerable amplification of seasonality of
streamflow (Juen et al., 2007; Vuille et al., 2008). Hinzman et al. (2020) report a widespread increase in nonlinearity of recessions in Northern Sweden due to climate warming. In addition, they find that this nonlinearity is significantly higher in warm winters than in cold winters.

In the southern TP, the Yarlung-Zangpo River basin (YRB) with decades of observations offers an opportunity to estimate variations of the recession characteristics of streamflow as YRB is a typical large river that drains in the high and cold area of
TP (Fig. 1a). Recent studies have shown that the climate in the YRB has become warmer and wetter from 1980-2015 (Wang et al., 2021). Climate warming has reduced the buffering effect of glacial and permafrost on streamflow, leading to catchment property change with shorter streamflow response time to precipitation in YRB (Wang et al., 2021). These changes must have affected streamflow recession characteristics.

The objective of this study is to investigate temporal and spatial variations of streamflow recession characteristics driven by
climate and landscape changes in YRB. The changes of these characteristics are examined using comparisons and contrasts of streamflow recessions in different sub-basins and time periods. In order to describe temporal variability of the recession characteristics under climate warming, the recession parameters of $a$ and $b$ are fitted by individual recessions from 1980 to 2015 and then regressed with mean temperature in recession period for each sub-basin of YRB. Sensitivity analysis shows the effect of climate warming on the recession parameters, the recession rates, and the storage/recharge-discharge relations in
different sub-basins. They show the extent of the nonlinearity in the variation of the streamflow recessions in this glaciated basin of TP.



## 2. Study region and data

The YRB (28.2°-31.2°N; 82.0°-94.9°E) is the largest river basin in TP (Fig. 1). The main stem of YRB is formed by major suture zones in southern TP, resulted from the collision between the Indian plate and the Eurasian plate. The modern YRB flows along the suture from the west to the east before bending to the south at the eastern Himalayan syntaxes with an average gradient about 2.63 ‰ (Fig. 1a) (Tan et al., 2021). In this study, we selected the upstream of the Great Gorge of YRB (main stem about 1100 km with an area of $2.0 \times 10^5$ km$^2$). The elevation of the study area drops drastically from 6234 m in the west to 2030 m in the east (You et al. 2007, Wang et al., 2021).

Climate in YRB is heavily influenced by the Indian monsoon in summer and the westerlies in winter (Ren et al., 2018; Tian et al., 2020). From the west to the east of the basin, mean annual temperature varies from -9.3 to 22.0 °C, and the mean annual precipitation varies from 300 to 1050 mm. Nearly 90% of annual precipitation falls during June to September. As a result, the mean annual total streamflow of the entire basin, 289.7 mm, is highly unevenly distributed in seasons. The summer streamflow is derived from monsoon rainfall and glacier meltwater. Groundwater accounts for about 54% of the annual streamflow.

There are four hydrological stations along the main stem of YRB: LZ, NGS, YC, and NX shown in Fig. 1a. Two additional hydrological stations, YBJ and LS, are located in the major tributaries of the Lhasa River which originates from the Nyainqêntanglha Mountains north of YRB (Fig. 1a). Daily streamflow data from 1980 to 2015 are available at these hydrological stations, except LZ. Accordingly, we divide YRB into five sub-basins, with three nested sub-basins of NGS, YC, and NX in the main stem of YRB, and two sub-basins of YBJ and LS in the tributary of the Lhasa River.

There are four main dams/reservoirs (marked by the purple squares in Fig. 1a) in YRB above the NX hydrological station. The reservoirs, ML, ZK, PD, and ZM, were built in 1999, 2007, 2013, and 2014, respectively. The reservoirs of ML, PD, and ZM are operated daily while the reservoir of ZK is operated seasonally. The impact of reservoir regulations on streamflow is minor for the sub-basins of NGS, YC, and NX in the main stem of YRB because the reservoirs are daily operated and affect less than 10% of the areas in the tributaries. In the tributary of the Lhasa River, the sub-basin YBJ has no reservoirs while the sub-basin LS has two reservoirs, PD and ZK, which have impacts on streamflow.

Daily gridded data (0.1°×0.1 °spatial resolution) of precipitation (P), and mean surface air temperature (T) during 1980–2015 were provided by the National Tibetan Plateau Data Center (Yang and He, 2019; He et al., 2020; http://data.tpdc.ac.cn). The sub-basin averaged P and T are calculated by the geometric mean of the gridded data.



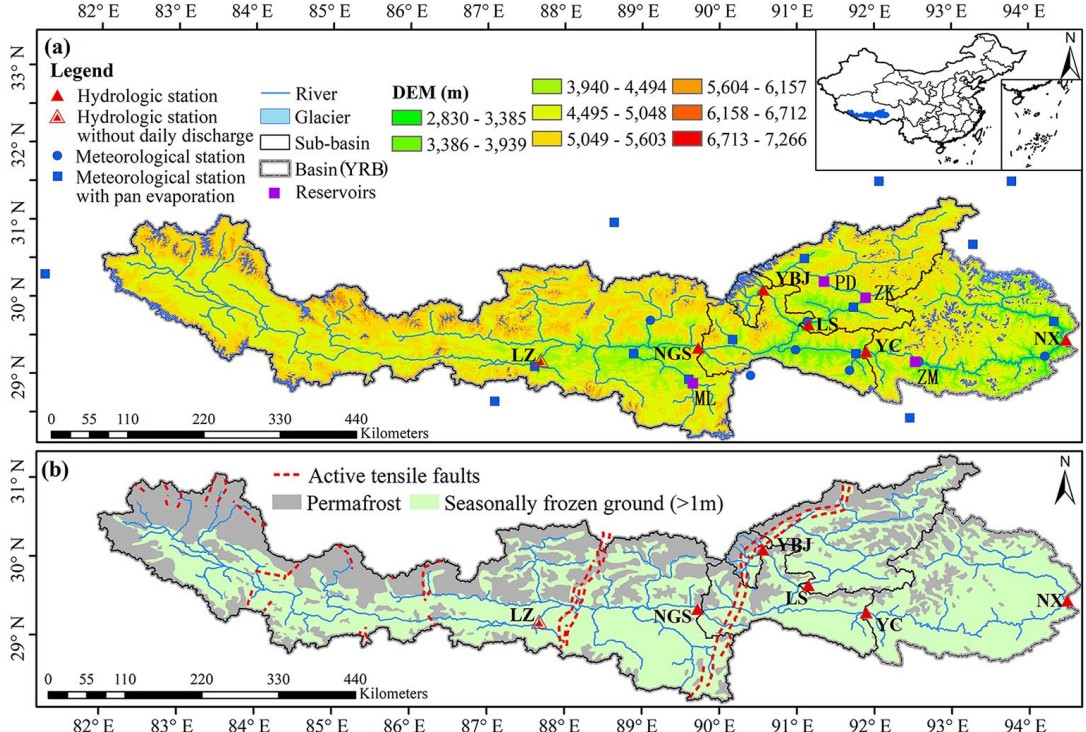

**Figure 1.** (a): Location of the Yarlung-Zangpo River basin (YRB, the entire basin above NX hydrological station) and its five
sub-basins (NGS, YC, NX, YBJ, and LS) from upstream to downstream (NGS for Nugesha, YC for Yangcun, NX for Nuxia,
YBJ for Yangbajain, and LS for Lhasa). (b): Distribution of permafrost and seasonally frozen ground in 2012 and active tensile
faults (the red dotted lines).

The glacier area, permafrost area, and the normalized difference vegetation index (NDVI) were collected from National Tibetan
Plateau Data Center (Table 1, http://data.tpdc.ac.cn). Glacier areas are located at altitudes from 3370 to 6460 m above the sea
level (Yao et al., 2010, Fig. 1a). The glacier and permafrost area before 2000 accounts for 1.88% and 41.8% of the YRB area,
respectively. These coverage percentages have reduced prominently since 2000 (Table 2). The annual mean NDVI was
calculated by the maximum value synthesis method from the Global GIMMS NDVI3g v1 dataset with a 15-day temporal
resolution and 1/12° spatial resolution. The vegetation types are mainly alpine meadow, alpine steppe in the upstream (LZ),
alpine shrubs, grasslands in the middle region (LZ-NGS), and alpine grassland and forest in the lower YRB (NGS-NX) (Liu
et al., 2014).

The annual depth of glacial melt data (G) during 1980–2015 are available from estimations using the degree-day model (Su et
al., 2015; Liu and Zhang, 2018). The details of the calculation procedures are provided in Wang et al. (2021). The annual ALT
is estimated using a linear statistical function of air freezing index ($FI_{air}$) described in Xu et al. (2017), in which $FI_{air}$ is
calculated according to the cumulative value of the daily mean temperature below 0 °C in a year.





After the end of warm season (June-September), there is little precipitation (Hayashi, 2020) in YRB, and the melting of snow and glacier is minor due to cold temperatures (<-5 °C, Fig. 3). Subsequently, the flow discharge recedes from September to February of the following year. In this study, we use the daily discharge ($Q_t$) in the recession period, which is defined from 1 October to 15 February of the following year, in our analysis of the recession process.

**Table 1.** Information of the data used in this study.

| Data | Period | Resolution | Source |
|---|---|---|---|
| Precipitation (P, mm) | 1980~2015 | 0.1 °×0.1 °, & Obs. stations, daily | National Tibetan Plateau Data Center; http://data.tpdc.ac.cn http://data.cma.cn |
| Mean Temperature (T, ℃) | | | |
| Evapotranspiration (E, mm) | | Obs. stations, daily | |
| Discharge (Q, mm) | | | |
| NDVI | 1982-2015 | 1/12 °×1/12 °, 15-days | http://data.tpdc.ac.cn |
| Glacial area | 1976, 2000, 2013 | 30m×30m, Annual | http://data.tpdc.ac.cn and China's second glacier catalogue data |
| | 2006~2011 (in 2009) | Mean annual | |
| Permafrost and Frozen ground | 1983-1996 (1988), 1997, 2003, 2012, 2017 | Mean annual | http://data.tpdc.ac.cn |
| Active layer thickness (ALT) | 1982-2015 | Annual | Calculated based on a linear function from Xu et al. (2017) |

## 3. Methodologies

### 3.1 Detection of changes in annual climate and hydrological series

The Mann-Kendall (MK) method (Mann, 1945; Kendall, 1975) combines trend-free pre-whitening treatment (TFPW-MK) (Yue & Wang, 2002) with Sen-slope (Sen 1968). The TFPW-MK is used in this study to test the temporal trend of annual

sequence for the meteorological and hydrological variables at the specified significance level of $\alpha$=0.05.

The Pettitt method is applied to detect the change point (year) of annual hydrological and meteorological variables. Pettitt method is nonparametric and has been widely used in mutation point detection (Pettitt, 1979; Mallakpour et al., 2016; Wang et al., 2021).

### 3.2 Streamflow recession analysis

Based on analytical solutions to the Boussinesq equation, the relationship of streamflow ($Q$) and streamflow change (-d$Q$/d$t$) in the recession period is expressed by the power-law equation (Brutsaert and Nieber, 1977):

$$dQ/dt = -aQ^b \qquad (1)$$



where $Q$ is streamflow (mm·d$^{-1}$), d$Q$/d$t$ is streamflow recession (mm·d$^{-2}$), $t$ is time (day), and $a$ (mm$^{1-b}$·d$^{b-2}$) and $b$ (dimensionless) are the recession coefficients (Brutsaert and Nieber, 1977; Tashie et al., 2020).

Based on (1), the relationship between dynamic groundwater storage ($S$) and streamflow ($Q$) can be derived:

$$S = KQ^m \tag{2}$$

where $K$ (=$[a \cdot (2-b)]^{-1}$ mm$^{b-1}$·d$^{2-b}$), and $m$ (=2-$b$).

The recession timescale ($\tau$) is estimated to measure the recession rates of individual recessions (Kirchner, 2009), and defined as

$$\tau = \frac{dS}{dQ} = \frac{1}{aQ^{b-1}} \tag{3}$$

From (1)-(3), the storage sensitivity of discharge ($\lambda_S$) for the recession curve (Berghuijs et al., 2016) is

$$\lambda_S = \frac{dQ/Q}{dS} = \frac{1}{\tau Q} = aQ^{b-2} \tag{4}$$

where $\lambda_S$ (mm$^{-1}$) is a measure of the sensitivity of instantaneous discharge values to water storage changes, and indicates the fractional increase in discharge for each unit of increase in storage. The larger (or smaller) the values of $a$ (or $b$) the more

sensitive the discharge to water storage.

Both the relationships of -d$Q$/d$t$ – $Q$ and $S$–$Q$ are linear if $b$=1 and nonlinear if $b \neq 1$. When $b \neq 1$, the discharge recession is

$$Q_t = Q_0(1 + Q_0^{b-1}a(b-1)t)^{1/(1-b)} \tag{5}$$

where $Q_0$ and $Q_t$ are initial discharge and discharge at time $t$. For any specific initial discharge $Q_0$, the larger the $b$ is the faster the hydrograph recession is for high discharge and the more stable the recession is for low discharge (Tashie et al., 2020).

The recession parameters $a$ and $b$ can be determined by fitting the daily observation data points of $(\Delta Q/\Delta t) - Q$ in log–log space using linear least-squares regression. The fitted parameter values of $a$ and $b$ are used to estimate -d$Q$/d$t$ and $Q_t$ using (1) and (5). The accuracy of the estimated -d$Q$/d$t$ values is evaluated by the root-mean-square logarithmic error (RMSLE) (Bekele and Nicklow, 2007):

$$\mathrm{RMSLE} = \left[\frac{1}{N}\sum_{i=1}^{N}\left(\log(-dQ_{est}(i)/dt) - \log(-dQ_{obs}(i)/dt)\right)^2\right]^{1/2} \tag{6}$$

Where $Q_{obs,i}$ and $Q_{est,i}$ are the observed and estimated discharges, respectively. $dQ_{est}(i)/dt$ and $dQ_{obs}(i)/dt$ are derivatives of (1). In practice, $\Delta Q/\Delta t$ is determined from the observed recession segments $\Delta Q$ in time interval $\Delta t$. $N$ in (6) is the number of data points of -d$Q$/d$t$ in individual recessions.

For each recession hydrograph, the fitting should ensure that the estimated volume of recession discharge approaches to that of the observed recession during the study period (Dralle et al., 2017), or by minimizing their differences

$$E_{MAP} = \frac{1}{N}\sum_{i=1}^{N}\left|\frac{Q_{obs,i}-Q_{est,i}}{Q_{obs,i}}\right| \tag{7}$$





where $E_{MAP}$ is the absolute relative error between $Q_{obs.i}$ and $Q_{est.i}$ over the recession period.

### 3.3 Changes of recession characteristics under climate warming

Recession coefficients $a$ and $b$ are functions of catchment properties, such as the hydraulic conductivity, the drainage density of the basin, and drainable porosity. In cold regions, recession coefficients are closely related to the thickness of the active layer in the soil profile above permafrost layer (Bense et al., 2012; Brutsaert and Hiyama, 2012). Changes of these catchment properties depend on daily, seasonal, and annual temperature variability in the refreezing areas. For example, the transition from unfrozen to frozen ground for temperature varying between 0 and -0.5°C coincides with a reduction in hydraulic conductivity of several orders of magnitude for saturated porous media (Burt and Williams, 1976; McCauley et al., 2002). On the other hand, when temperature rises, the thawing front in the active soil layer moves progressively downward as summer proceeds, leading to increasing water storage in the active layer. If the frozen soil beneath the thawing front is ice-saturated (thus relatively impermeable), the active soil layer can function as a very shallow perched aquifer that controls streamflow response to snowmelt and summer precipitation (Carey and Woo, 2005; Yamazaki et al., 2006; Wright et al., 2009; Koch et al., 2014).

Accordingly, the variability of the parameters $a$ and $b$ can be expressed as a function of temperature ($T$), i.e., $a(T)$ and $b(T)$. In this study, we use

$$\begin{cases} a(T) = \alpha \cdot \exp(\alpha_1 T) \\ b(T) = \beta \cdot \exp(\beta_1 T) \end{cases} \tag{8}$$

where $\alpha$ and $\alpha_1$ are coefficients for $a(T)$, $\beta$ and $\beta_1$ are coefficients for $b(T)$, and $T$ is the mean surface air temperature in the recession period ($T_{re}$). These coefficients can be obtained by fitting the recession parameters of individual recession events with known $T_{re}$ in each sub-basin.

From (2), the temporal change of $S$ ($\Delta S$) can be expressed

$$\Delta S = \frac{\partial S}{\partial K}\Delta K + \frac{\partial S}{\partial m}\Delta m + \frac{\partial S}{\partial Q}\Delta Q \tag{9}$$

Incorporating $K=1/[a \cdot (2-b)]$ and $m=2-b$, we can write $\Delta S$ as

$$\Delta S = \frac{\partial S}{\partial K}\frac{\partial K}{\partial a}\Delta a + \left(\frac{\partial S}{\partial K}\frac{\partial K}{\partial b} + \frac{\partial S}{\partial m}\frac{\partial m}{\partial b}\right)\Delta b + \frac{\partial S}{\partial Q}\Delta Q \tag{10}$$

or

$$\Delta S = \lambda_a \Delta a + \lambda_b \Delta b + \lambda_Q \Delta Q \tag{11}$$

where $\lambda_a$, $\lambda_b$, and $\lambda_Q$ are the sensitivity coefficients of $\Delta S$ to $a$, $b$, and $Q$, respectively, and can be derived as follows





$$\begin{cases} \lambda_a = \frac{\partial S}{\partial K}\frac{\partial K}{\partial a} = \frac{-1}{a^2(2-b)}Q^{2-b} \\ \lambda_b = \frac{\partial S}{\partial K}\frac{\partial K}{\partial b} + \frac{\partial S}{\partial m}\frac{\partial m}{\partial b} = \frac{1}{a(2-b)}Q^{2-b}\left(\frac{1}{2-b} - \ln Q\right) \\ \lambda_Q = \frac{\partial S}{\partial Q} = \frac{1}{aQ^{b-1}} = \tau \end{cases} \qquad (12)$$

Because the recession parameters $a$ and $b$ are functions of $T$ in (8), $S$ is a function of $T$ and $Q$. The change of $S$ ($\Delta S$) can therefore be expressed as

$$\Delta S(T) = \frac{\partial S}{\partial T}\Delta T + \frac{\partial S}{\partial Q}\Delta Q \qquad (13)$$

or

$$\Delta S(T) = \lambda_T \Delta T + \lambda_Q \Delta Q \qquad (14)$$

where $\lambda_T$ is the sensitivity coefficient of $\Delta S$ to $T$. $\lambda_T$ can be derived as

$$\lambda_T = \frac{\partial S}{\partial T} = \frac{1}{a(2-b)}Q^{2-b}\left(\frac{b(\beta_1+\alpha_1)-2\alpha_1}{2-b} - b\beta_1 \ln Q\right) \qquad (15)$$

## 4. Results

### 4.1 Spatial and temporal variations of climate and hydrological variables

#### 4.1.1 Spatial variations

According to the mean values of the observed climate and hydrological variables during 1980–2015 in different sub-basins, climate has become warmer and wetter (Fig. 2a–2c and Table 2) along the main stem of YRB. The wet trend is largest in YBJ, as well as LS sub-basins, whereas the mean temperature of YBJ is lowest because most of its area is at high altitudes. The percentage of glacier area is 1.63%, 1.52%, and 1.92% in NGS, YC, and NX sub-basin, respectively, and 9.91% and 0.75% for YBJ and LS, respectively. The percentage of permafrost area (PPA) ranges from 41.8%–47.7% in the five sub-basins.

The mean annual streamflow and streamflow in the recession period (in units of mm) increased (Figs. 2d and 2e), resulting in higher runoff coefficient $Rc$ ($R/P$) towards the wetter downstream of the main stem of YRB. However, the wettest sub-basins of YBJ and LS do not have the greatest discharge and high $Rc$, possibly because of the icy environment. Daily coefficient of variation (CV) of streamflow is higher in YBJ and LS and in upstream of NGS. CV decreases towards the wetter downstream of YRB, partially because of strengthening watershed regulation as the sub-basin areas increase and the dams are included in the area of the analysis.





**Table 2.** Summary of sub-basin characteristics in YR basin.

| Regions | | NGS | YC | NX | YBJ | LS |
|---|---|---|---|---|---|---|
| Drainage area ($10^4$ km$^2$) | | 10.86 | 16.51 | 20.32 | 0.31 | 3.06 |
| Mean Elevation (m asl) | | 3776 | 3553 | 2944 | 4255 | 3794 |
| Glacier area (km$^2$) | (year) 1976 | 2070 | 2902 | 4285 | 241 | 283 |
| | 2001 | 1822 | 2562 | 3821 | 227 | 257 |
| | 2009* | 1674 | 2355 | 3782 | 224 | 255 |
| | 2013 | 1489 | 2217 | 3709 | 220 | 247 |
| Percentage of permafrost area (%, 2003) | | 47.7 | 44.1 | 41.8 | 44.9 | 44.5 |
| Mean annual value | $P$ (mm) | 354 | 386 | 426 | 433 | 536 |
| | $T$ ($^0$C) | -1.51 | -1.16 | -0.99 | -2.15 | -1.91 |
| | $T_{re}$ ($^0$C) | -6.23 | -5.95 | -5.73 | -8.37 | -6.83 |
| | $Q$ (mm) | 144 | 180 | 290 | 240 | 302 |
| | $Q_{re}$ (mm) | 26.5 | 34.0 | 52.3 | 31.5 | 47.9 |
| | $Rc$ (R/P) | 0.410 | 0.460 | 0.680 | 0.555 | 0.563 |
| | $CV_1$ | 1.099 | 1.061 | 0.931 | 1.138 | 1.123 |
| | $CV_2$ | 1.155 | 1.070 | 0.970 | 1.106 | 1.203 |

Note: * Glacier area is from China's second glacier catalogue data in 2009. The mean of $A_{ice}$ in year 1976 and 2001 is used as a reference value in the sub-period before 1997, and the mean of $A_{ice}$ in 2001, 2009, and 2013 is used as a reference value in the sub-period after 1997. The subscript "1", and "2" represent 1980-1996, and 1997-2015, respectively.

### 4.1.2 Annual variations of climate and hydrological variables during 1980–2015

Figures 2a–2f show annual variations of the observed climate and hydrological variables from 1980 to 2015. Tested by TFPW-
MK, annual mean temperature T and temperature in the recession period $T_{re}$ rose significantly ($p < 0.05$) in all sub-basins of YRB. Annual T rose at a rate of 0.045–0.075℃·a$^{-1}$ for the five sub-basins, smaller than the rate of 0.070–0.097 ℃·a$^{-1}$ for annual $T_{re}$. Annual precipitation P increased, tested to be significant ($p < 0.1$), in the sub-basins of YC, NX, and LS, but insignificant in NGS and YBJ. Annual glacier meltwater G increased significantly in all sub-basins. The rate of G increase is from 0.46 mm·a$^{-1}$ in LS to 2.86 mm·a$^{-1}$ in YBJ. Under the warmer and wetter climate, vegetation coverage trends to increase.
NDVI increases at a rate of 5.1$^{-4}$–8.03$^{-4}$ a$^{-1}$ during 1980–2015 in the four sub-basins YC, NX, and NGS and YBJ, but decreases at a rate of -1.03$^{-4}$ a$^{-1}$ in LS (Fig. 2g).

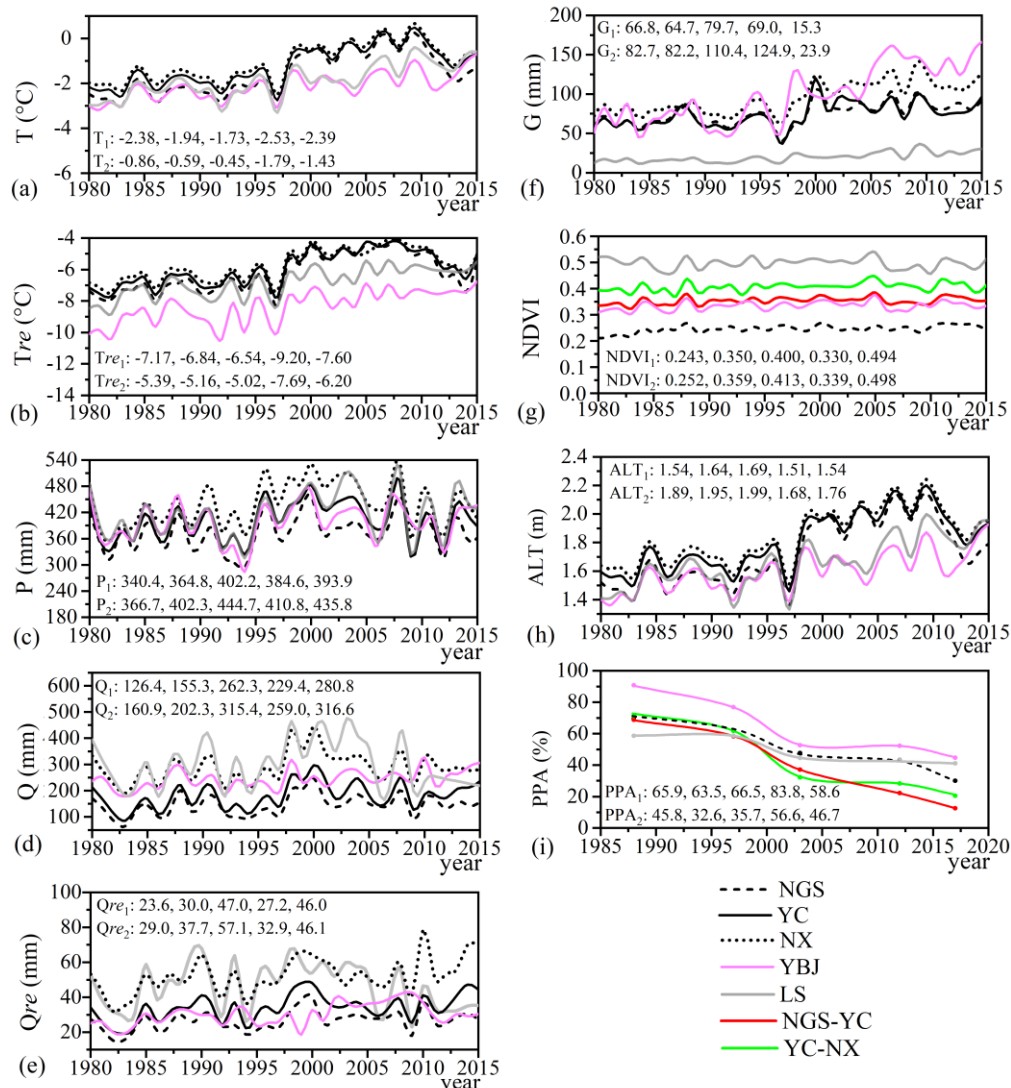

**Figure 2.** Variations of (a) annul mean temperature, (b) mean temperature in recession period (Tre), (c) precipitation, (d) discharge (Q), (e) discharge in recession period (Qre), (f) glacier meltwater (G), (g) NDVI, (h) active layer thickness (ALT),

and (i) percentage of permafrost area (PPA) from 1980 to 2015 in the five sub-basins. The values listed in each panel represent mean values of NGS, YC (or NGS-YC), NX (or YC-NX), YBJ and LS in sequence. The subscripts "1" and "2" refer to the early period from 1980 to 1996 and the recent period from 1997 to 2015, respectively. NGS-YC refers to the area between the nested sub-basins NGS and YC, and YC-NX is for the area of the nested sub-basins YC and NX.

Over the same period, the annual mean discharge Q and the mean discharge in the recession period $Q_{re}$ increased significantly,

except for $Q_{re}$ in LS sub-basin. The annual Q increased at a rate of 1.16–1.68 mm·a$^{-1}$, higher than the rate of 0.22–0.47 mm·a$^{-1}$





for $Q_{re}$. In contrast, annual $Q_{re}$ in LS decreased insignificantly (Fig. 2d), possibly because of initial water storage when the ZK reservoir (Fig. 1a) began operation around 2007.

In Wang et al. (2021), a point of dramatic change was detected around 1997 by the Pettitt' test for annual T, G, and Q, and around 1995 for annual P in the sub-basins of NGS, YC and NX. In this study, the same point of change in 1997 was detected in the annual series of $T_{re}$ and $Q_{re}$ in NGS, YC and NX sub-basins, and in the annual series of T, G, and Q in YBJ and LS sub-basins. The annual time series of Q in LS was detected to have two additional points of change in 1995 and 2005, attributable to the increased impact of human (reservoir operation) activities (Cai et al., 2021). The point of change in 1997 has also been identified by the dramatic changes of surface conditions, i.e., the reversed NDVI trend before and after 1997 (Fig. 2g), increased ALT after 1997 (Fig. 2h), and accelerated thawing of permafrost after 1997 (Fig. 2i). Accordingly, in this study, the study period of 1980–2015 is separated into two periods: the early period from 1980 to 1996, and the recent period from 1997 to 2015.

Compared with P and T in the two periods, climate in the recent period changes to be markedly warmer and wetter. The mean annual *P* after 1997 increased by 27–46 mm or 7.9–10.7 % more than that in the early period for the five sub-basins. The mean annual T in the recent period is 0.75–1.52℃ warmer than that in the early period, and a larger rise of 1.40–1.78℃ was found for the mean $T_{re}$ after 1997. These changes concurred with 8.6–55.9 mm or 23.8–81.1% increase of glacier meltwater after 1997 in the five sub-basins. As a result, mean annual streamflow increased by 29.6–50.2 mm or 12.7–31.5% in the recent period compared to that before 1997. This increase in Q and $Q_{re}$ after 1997 is much larger in the upstream sub-basins such as NGS and YC.

### 4.1.3 Annual recession characteristics

As shown in Fig. 3, the annual hydrographs in the five sub-basins are consistent, delineating a single peak response to maximum precipitation and temperature in July–August. The statistic values of daily discharge series in the two periods show that the mean value in the recent period exceeds that in the early period in all sub-basins. Meanwhile, daily discharge variability in the recession of the annual hydrograph in the recent period is also greater than that of the early period, shown by larger coefficients of variation (*CV*) after 1997 for most sub-basins, except the glacierized sub-basin YBJ (Table 2).

The recessions in the sub-basins are faster in the recent periods when the climate is warmer and wetter. As shown in Figs. 4a-4e, the fitted line of the -d$Q$/d$t$ vs. Q after 1997 has a steeper slope. According to the non-overlapping moving averages of the 5-day series of the recession discharge, the estimated average recession rate after 1997 [$(\Delta Q/\Delta t)_2$] is larger than that before 1997 [$(\Delta Q/\Delta t)_1$] as indicated by the positive values of $\Delta v_Q = (\Delta Q/\Delta t)_2 - (\Delta Q/\Delta t)_1$ in Fig. 4f.





**Figure 3.** (a)-(e): Mean daily precipitation *P*, temperature *T*, and discharge *Q* in a hydrological year (from 1 March to 28 February of the following year) for the two periods in the five sub-basins. The red dashed rectangle in (a) shows the hydrograph recession from 1 October to 15 February of the following year, and the shading shows the range of daily variability of *P*, *T*, and *Q* in each period.





### 4.2 Estimation of the recession parameters

Targeting the observed hydrograph in each year, its recession was fitted to obtain the recession parameters $a$ and $b$. The results are summarized in Table 3. The mean annual value of parameter $a$ during 1980–2015 ranges from 0.022 to 0.042 mm$^{1-b}$ d$^{b-2}$, and the mean annual value of $b$ ranges from 1.36 to 1.85 for the five sub-basins. The mean values of $a$ and $b$ decrease from upstream to downstream along the main stem of YRB (Table 3). Figure 5 shows the errors of the estimated recession (RMLSE and $E_{MAP}$) for the recession curve of each sub-basin. Mean annual RMLSE is less than 0.15 in all sub-basins. The mean annual

$E_{MAP}$ is lower than 10%, except in the sub-basins YBJ and LS where $E_{MAP}$ is 0.15 and 0.14, respectively.

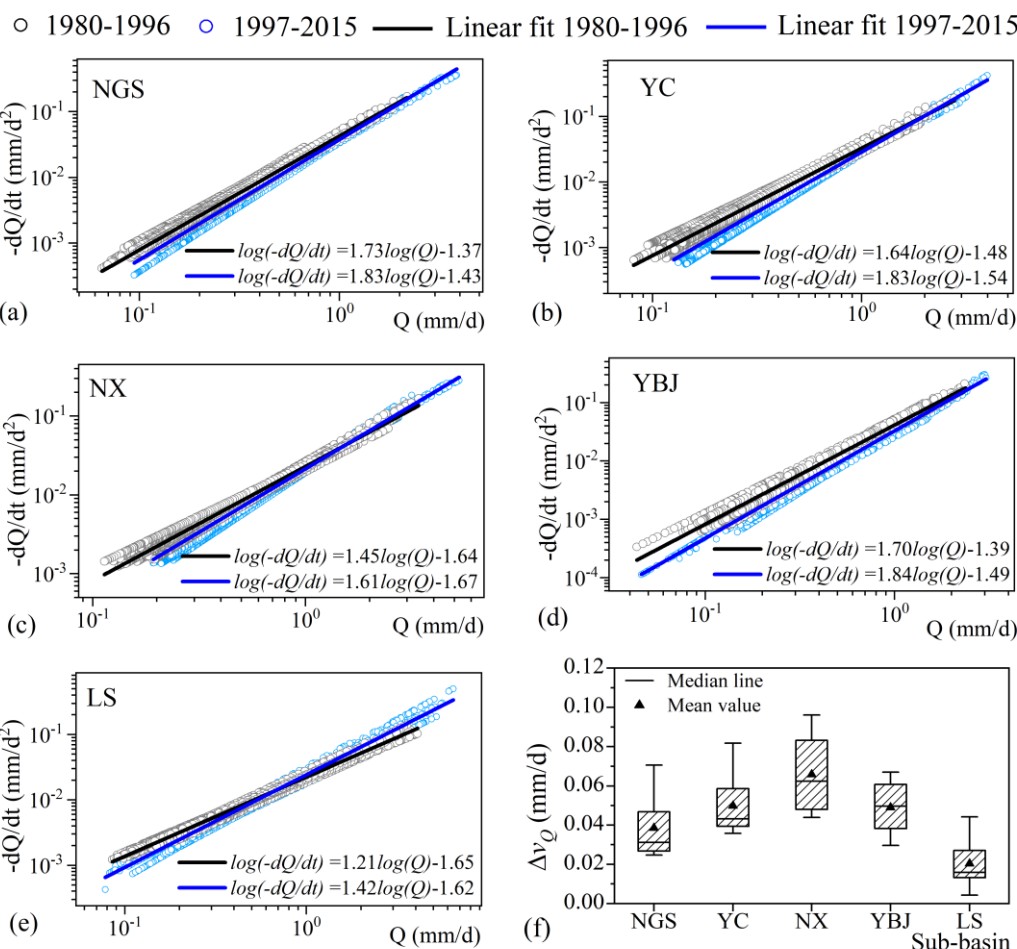

**Figure 4.** (a)-(e): Plot of -d$Q$/d$t$ vs. $Q$ in log-log space for each recession hydrograph during 1980–2015, and the fitting lines [log(-d$Q$/d$t$) = $b$log($Q$)+log($a$)] for the data points in the two periods for the five sub-basins. (f): Differences of mean recession rates between the two periods ($\Delta v_Q$) estimated from the non-overlapping moving averages of the 5-days' series.



Figure 6 shows the relationship of annual value and 4–year moving average value of the recession parameters $a$ and $b$ with mean surface air temperature in the recession period ($T_{re}$) in each sub-basin. The exponential function between $a$ or $b$ and $T_{re}$ is fitted with a high determination coefficient ($R^2$ ranges 0.63–0.81 for $a$ and 0.58–0.87 for $b$). These results show that $a$ decreases exponentially with increasing $T_{re}$ for the sub-basins, except LS, while the $b$ value increases exponentially with increasing $T_{re}$ in all sub-basins.

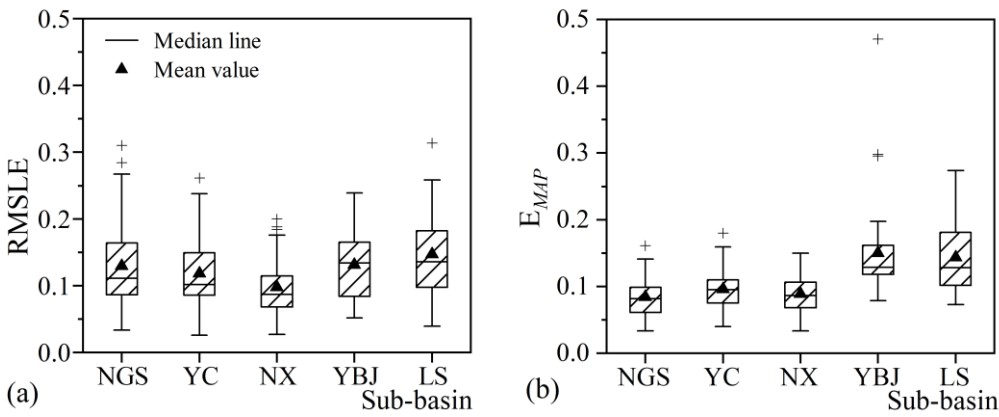

**Figure 5.** The Box-plot of (a) estimated error of -dQ/dt (RMSLE) between observed and estimated discharge; and (b) the absolute relative error ($E_{MAP}$) between annual mean observed and estimated discharge for individual recession hydrographs in each sub-basin.

For the multiyear mean values of parameters $a$ and b in the two periods (Table 3), the mean value of $a$ in the recent period
ranges between 0.021–0.039 mm$^{1-b}$ d$^{b-2}$, smaller than that in the early period when they range between 0.022–0.046 mm$^{1-b}$ d$^{b-2}$ in the sub-basins, except LS. Oppositely, the mean value of $b$ in the recent period ranges between 1.47–1.89, larger than that in the early period (1.25–1.81) for all sub-basins. These results indicate that warming and wetting climate increases the nonlinearity of the recession ($b$) and reduces streamflow stability [log($a$)] for most sub-basins in YRB.

### *4.3 Change of storage-discharge relationship under climate warming*

The strong sensitivity of the recession parameters of $a$ and $b$ to the near surface air temperature means that climate warming can change the nonlinear relationship between the water storage ($S$) and discharge/streamflow ($Q$) (Eq. (2)). This change is demonstrated by the increase of the recession coefficient $K$ in the recent period (11.5–73.2 mm$^{b-1}$·d$^{2-b}$ for $\Delta K$ in Table 3) and decrease of $m$ (=2-$b$) in Eq. (2) because $b$ increases with the temperature (Table 3 and Fig. 6). The increase of $K$ and decrease of $m$ mean a lower discharge for a specific storage in the recent warmer period. As an example, we show in Fig. 7 the
relationship of $S$ with $Q$ for different $K$ and/or $m$ between the two periods in sub-basin YC. $Q$ decreases significantly with increase of $K$ or decrease of $m$ for any specific storage $S$ (Figs. 7a and 7b). The combination of increase of $K$ and decrease of $m$ leads to marked decrease of $Q$ for any specific storage $S$ (Fig. 7c). Correspondingly, the recession timescale ($\tau$=d$S$/d$Q$)



increases by 5.2–20.7 days in the recent warmer period in all sub-basins (Table 3), especially the glaciated sub-basin YBJ. The

increase of $\tau$ means increase in storage for any specific $Q$ because $S(Q)-S_0 = \int_{S_0}^{S} dS = -\int_{Q_0}^{Q} \tau(Q)dQ > 0$ in the recession.

**Table 3.** Mean annual parameters of $a$ (mm$^{1-b}$ d$^{b-2}$), $b$ (-), recession coefficient $K$ (mm$^{b-1}$ d$^{2-b}$) and recession timescale $\tau$ (d). * is significant tested by TFPW-MK ($p < 0.05$). The values in the bracket refer to the range of annual value.

| Period | Index | Mean annual value | | | | |
|---|---|---|---|---|---|---|
| | | NGS | YC | NX | YBJ | LS |
| 1980–2015 | $a$ | 0.042 (0.033~0.060) | 0.032 (0.025~0.043) | 0.022 (0.019~0.027) | 0.038 (0.025~0.052) | 0.024 (0.018~0.032) |
| 1980–1996 | | 0.046 | 0.035 | 0.023 | 0.043 | 0.022 |
| 1997–2015 | | 0.039 | 0.029 | 0.021 | 0.034 | 0.025 |
| $\Delta a$ | | **-0.007*** | **-0.006*** | **-0.002*** | **-0.009*** | **0.003*** |
| 1980–2015 | $b$ | 1.85 (1.645~1.990) | 1.70 (1.506~1.992) | 1.54 (1.297~1.789) | 1.85 (1.607~1.979) | 1.36 (1.117~1.783) |
| 1980–1996 | | 1.81 | 1.67 | 1.48 | 1.78 | 1.25 |
| 1997–2015 | | 1.89 | 1.73 | 1.59 | 1.90 | 1.47 |
| $\Delta b$ | | **0.08*** | **0.06** | **0.11*** | **0.11*** | **0.22*** |
| 1980–2015 | $K$ | 129.7 (73.7~196.5) | 127.3 (68.3~203.0) | 97.9 (63.4~131.0) | 142.9 (59.0~205.4) | 64.2 (46.3~88.8) |
| 1980–1996 | | 100.9 | 94.0 | 86.6 | 98.9 | 58.0 |
| 1997–2015 | | 155.5 | 157.5 | 107.6 | 172.1 | 69.5 |
| $\Delta K$ | | **54.6*** | **63.5*** | **21.0*** | **73.2*** | **11.5*** |
| 1980–2015 | $\tau$ | 90.8 (65.7~117.3) | 89.7 (64.0~126.6) | 75.3 (56.7~108.4) | 93.0 (56.1~141.7) | 65.0 (49.0~110.9) |
| 1980–1996 | | 88.2 | 83.1 | 71.7 | 77.8 | 57.3 |
| 1997–2015 | | 93.4 | 96.4 | 78.8 | 98.5 | 71.8 |
| $\Delta \tau$ | | **5.2** | **13.4*** | **7.1** | **20.7*** | **14.5*** |

The lower discharge for a specific storage or higher storage for any specific discharge can be further illustrated by the results of the storage sensitivity of discharge ($\lambda_S$ in Eq. (4)) in Table 4. The mean value of $\lambda_S$ during 1980–2015 ranges between 0.036 –0.059 mm$^{-1}$ for the five sub-basins. These values mean that 1 mm decrease in storage results in 3.6–5.9% decrease in discharge.

In terms of Eq. (4), the increase of the recession timescale ($\tau$) and discharge ($Q$) should result in the decrease of $\lambda_S$, supported by the smaller $\lambda_S$ values in the recent warmer period (see the negative values of $\Delta\lambda_S$ in Table 4). Meanwhile, for the sub-basins in the main stem of YRB (NGS, YC and NX), the mean annual value of $\lambda_S$ during 1980–2015 decreases towards the warmer and wetter downstream (from NGS to NX, see Table 4). This change suggests that climate warming weakens $\lambda_S$, meaning that a unit decrease in storage releases less water to discharge in the recession period. This is especially so in glaciated basins, e.g.,

YBJ, where the decrease of $\lambda_S$ ($\Delta\lambda_S$ in Table 4) is largest in the recent period, corresponding to the largest increase of $\tau$. We note that the decrease of $\lambda_S$ ($\Delta\lambda_S$) in LS is relatively small in the recent period primarily because of regulation of reservoirs on discharge.





**Table 4.** The storage sensitivity of discharge ($\lambda_S$, mm$^{-1}$), and sensitivity coefficients of recession parameters of $a$ ($\lambda_a$) and $b$ ($\lambda_b$), $T$ ($\lambda_T$) and $Q$ ($\lambda_Q$) to storage change ($\Delta S$) during different periods for each sub-basin. The values in the bracket refer to the range of annual value.

| Period | Index | Mean annual value | | | | |
|--------|-------|------|------|------|------|------|
| | | NGS | YC | NX | YBJ | LS |
| 1980–2015 | $\lambda_S$ | 0.059 (0.035~0.095) | 0.048 (0.030~0.090) | 0.036 (0.026~0.058) | 0.053 (0.032~0.095) | 0.050 (0.035~0.085) |
| 1980–1996 | | 0.069 | 0.058 | 0.042 | 0.066 | 0.056 |
| 1997–2015 | | 0.050 | 0.042 | 0.031 | 0.041 | 0.046 |
| $\Delta\lambda_S$ | | **-0.019*** | **-0.016*** | **-0.012*** | **-0.025*** | **-0.010** |
| 1980–2015 | $\lambda_a$ | -2379 | -3040 | -3436 | -3193 | -2069 |
| 1980–1996 | | -1593 | -1814 | -2516 | -1872 | -1804 |
| 1997–2015 | | -3509 | -5125 | -4565 | -6036 | -2429 |
| $\Delta\lambda_a$ | | **-1917*** | **-3311*** | **-2049*** | **-4164*** | **-625** |
| 1980–2015 | $\lambda_b$ | 664 | 477 | 204 | 786 | 96 |
| 1980–1996 | | 435 | 270 | 157 | 371 | 72 |
| 1997–2015 | | 950 | 786 | 257 | 1428 | 129 |
| $\Delta\lambda_b$ | | **515*** | **515*** | **100*** | **1057*** | **58** |
| 1980–2015 | $\lambda_Q$ | 90.8 | 89.7 | 75.3 | 93.0 | 65.0 |
| 1980–1996 | | 88.2 | 83.1 | 71.7 | 77.8 | 57.3 |
| 1997–2015 | | 93.4 | 96.4 | 78.8 | 98.5 | 71.8 |
| $\Delta\lambda_Q$ | | **5.2** | **13.4*** | **7.1** | **20.7*** | **14.5*** |
| 1980–2015 | $\lambda_T$ | 37.1 | 51.0 | 25.1 | 95.2 | 11.9 |
| 1980–1996 | | 24.4 | 28.8 | 18.8 | 46.7 | 7.2 |
| 1997–2015 | | 52.5 | 83.8 | 32.4 | 163.5 | 18.8 |
| $\Delta\lambda_T$ | | **28.1*** | **55.1*** | **13.6** | **111.3*** | **11.7** |





**Figure 6.** (a)-(e): The exponential function of the recession intercepts *a* with mean air temperature in the recession period for each sub-basin. (f)-(j): Same as (a)-(e) but for recession slope *b*. The solid and open circles represent 4-year average and annual value, respectively.




### 4.4 Sensitivity of recession parameters to storage change under climate warming

Change of the sensitivity of discharge to storage ($\lambda_S$) should affect the recession processes, which is described by the sensitivity of the recession parameters to storage change $\Delta S$ ($\lambda_a$ and $\lambda_b$ for $a$ and $b$, respectively). As listed in Table 4, $\lambda_b$ shows positive while $\lambda_a$ has negative value in all five sub-basins. The larger negative $\lambda_a$ and positive $\lambda_b$ could be found in the recent period after 1997, particularly in the glaciated sub-basin YBJ ($\Delta\lambda_a$ and $\Delta\lambda_b$ in Table 4). Therefore, increase in storage due to climate warming will enhance the nonlinearity of recession ($b$) and weaken streamflow stability [$\log(a)$] in YRB. This sensitivity can be dampened however by anthropogenic effect (reservoir regulations) as suggested by the less sensitive result of the recession parameters of $a$ and $b$ to $\Delta S$ in LS sub-basin.

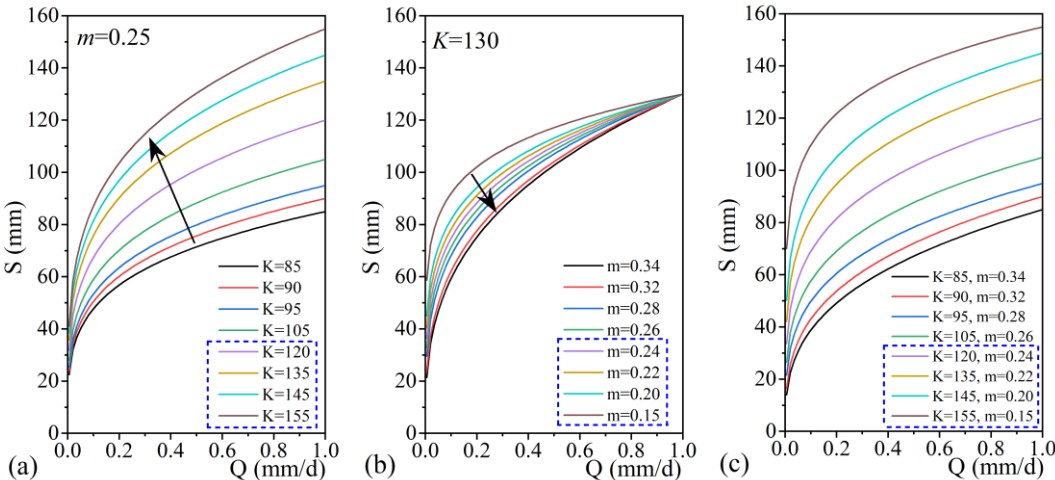

**Figure 7.** Relationship of storage $S$ and discharge $Q$ at different values of $K$ (a) and $m$ (b), and combinations of $K$ and $m$ (c) in the two periods for sub-basin YC. The different set of numbers in the blue dashed line box in each panel show the $K$ and $m$ values and therefore different $S$–$Q$ relationships in the recent period after 1997.

Effects of increase in storage on change of the streamflow recession characteristics can be illustrated by the storage change $\Delta S$ in response to changes in $T$ and $Q$ ($\Delta T$ and $\Delta Q$, respectively) in terms of Eq. (14). The values of $\lambda_T$ in Table 4 show an increase in the recent period in all sub-basins, especially in YBJ and YC. Thus, the enlarged storage is largely attributed to climate warming. As expected, $\lambda_T$ is smaller in the warm and wet NX sub-basin and the sub-basin LS with reservoir regulation. The values of $\lambda_Q$ in Table 4 also become bigger in the recent period in all sub-basins. These changes indicate that the climate warming increases storage and discharge.

However, the increase of discharge $\Delta Q$ in response to the increase in storage $\Delta S$ can be quite different in response to different rate of change of temperature $\Delta T$ in the five sub-basins. In terms of Eq. (14), the relationship between $\Delta S$ and $\Delta Q$ for different $\Delta T$ in the five sub-basins is shown in Fig. 8. As temperature rises, $\Delta S$ becomes greater while the greater increase of storage volume (in thawing soil layers) allows smaller amount of water to be released as baseflow. For example, as temperature rises





to be 1.2°C higher than mean annual temperature (i.e., $\Delta T = 1.2°C$), an increase of discharge (e.g., $\Delta Q = 0.2$ mm d$^{-1}$) corresponds to a storage increase of about 92 mm in NGS, 160 mm in YC, 63 mm in NX, 42 mm in LS and huge increase of

478 mm in YBJ (see the value of the $\Delta S$ vs. $\Delta Q$ point in Fig 8). These results suggest that a larger increase of water storage caused a smaller increase of baseflow in the glaciated sub-basins, reflecting a buffering effect of freezing on streamflow dynamics.

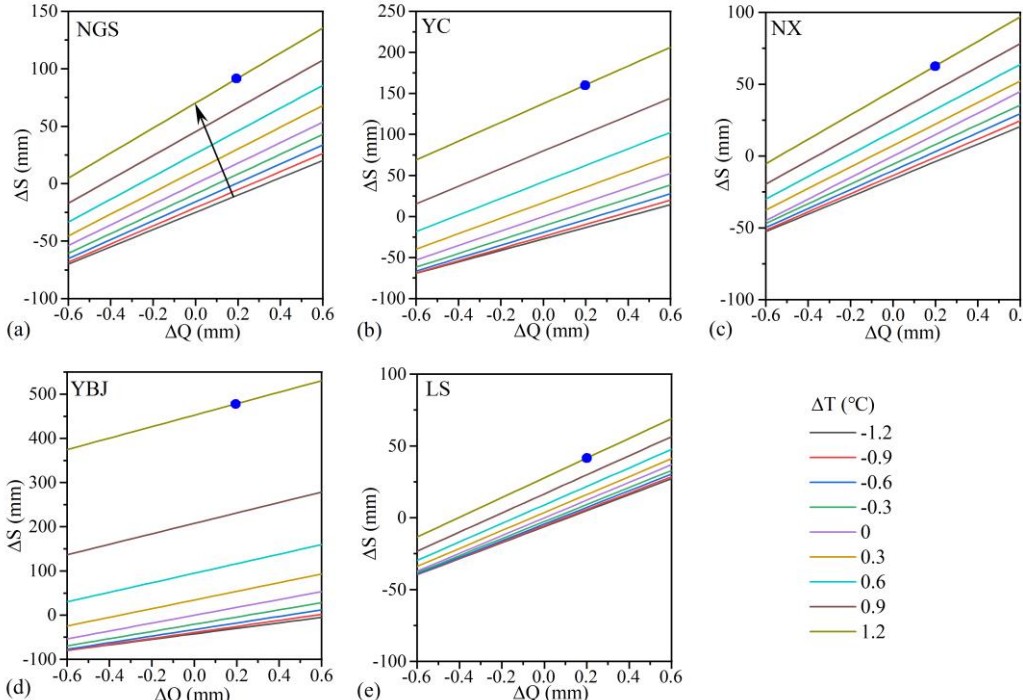

**Figure 8.** Changes in storage $\Delta S$ in relation to changes in discharge $\Delta Q$ under different changes in temperature $\Delta T$ for each

sub-basin. The changes in temperature $\Delta T$ refer to annual values relative to mean annual temperature in the recession period during 1980–2015. The solid circle refers the point of $\Delta S$ in response to 0.2mm of $\Delta Q$.

When the mean surface air temperature ($T_{re}$) increased from the early to the recent period, i.e., from -7.17 to -5.39 °C in NGS, -6.84 to -5.16 °C in YC, -6,54 to -5.02 °C in NX, -9.20 to -7.69 °C in YBJ, and -7.60 to -6.20 °C in LS, the estimated water storage in terms of $\lambda_T \Delta T$ in Eq. (14) increased between 15.2–132.6 mm for the five sub-basins. These increases contribute to

about 86.4–99.9% of the total increase in storage ($\lambda_T \Delta T / \Delta S$) in those sub-basins. They only cause 0.1–13.6% increase of discharge in terms of $\lambda_Q \Delta Q / \Delta S$ in Eq. (14). Moreover, this relationship varies among the five sub-basins with different glaciate conditions. In the warm and wet sub-basin NX with low glacial coverage, the increase in storage ($\lambda_T \Delta T$) is relatively small (35.4 mm and 86.4% of the total increase in storage) and the increase in discharge ($\lambda_Q \Delta Q$) is relatively large (5.56 mm and 13.6% of the total increase in storage). In the cold sub-basins with high glacial coverage, climate warming causes large increase





of storage but small increase of discharge. As an example, the YBJ sub-basin has 97.3% of the total increase in storage vs. only 2.7% of the total increase in discharge. Again, this relationship is distorted in basins with human regulatory actions in water management. In the sub-basin LS with strong reservoir regulations, changes in both the storage and discharge are small (e.g., 0.1% in the total increase of storage).

**5. Discussions**

Observations have shown that climate warming has accelerated glacier melting and permafrost thawing in cold climate and high-altitude regions. Subsequent changes are found in vegetation growth and thickening of talik and active soil layer thickness. These changes have altered land surface conditions and unconsolidated soil profiles and subsurface permafrost and redefined surface and groundwater exchange and balance in those regions (Fig. 9). Our case study of YRB in southern TP shows that accelerated glacier melting and permafrost thawing in YRB during 1980–2015 have substantially increased its dynamic

groundwater storage, defined as $S(Q) - S_0 = -\int_{Q_0}^{Q} \tau(Q)dQ$. These results with the decrease of terrestrial water storage (TWS) in south TP, including YRB (Wang et al., 2020) in recent decades, indicate that a transform of water storage from the solid form (glacier and permafrost) to the liquid volume (soil moisture, surface water in rivers/lakes, and groundwater) (Fig. 9b). According to water balance in a catchment, i.e., $dS/dt = (P_r - E - Q)$, where $E$ is evapotranspiration, $S$ is regarded as the "liquid volume" (here, change of $S$ is equal to the sum of changes in soil moisture and groundwater) and $P_r$ is the recharge from glacier

melting, permafrost thawing and precipitation, the increase of $S$ infers that $P_r$ is larger than the sum of $E$ and $Q$ in a study region. Because cold regions tend to have a greater coverage percentage of glacier and permafrost, glacier melting and permafrost thawing could substantially increase water storage under climate warming. Higher water storage could extend the recession period and sustain healthy annual streamflow.

Our study also shows that the increase of water storage and its effect on the annual recession of streamflow weakened towards

the warmer downstream areas of YRB (with diminishing glacier melting and permafrost thawing effect). Accordingly, if the climate warming continues, the shrinking of glacier and permafrost volume could eventually reach a point when there is not enough melting to recharge the liquid volume of water in YRB. From that point onward, steady streamflow in YRB would be in danger.

While the processes initiated by the accelerated glacier melting and permafrost thawing lengthen subsurface flow paths

(Hinzman et al., 2020) and the streamflow recession time ($\tau$), the increase of surface temperature and $E$ can also increase surface water loss. According to the discharge relation $-dQ/dt = -\frac{dQ}{dS}\frac{dS}{dt} = (-P_r + E + Q)/\tau$ (Kirchner, 2009), where $P_r$ can be neglected in the recession period, a faster recession ($-dQ/dt$) could occur under climate warming from faster decrease of storage ($dS/dt$) due to increasing of $E$. Meanwhile, the decreased sensitivity of discharge to the storage ($dQ/dS = 1/\tau$) as the storage expands in the warming climate would slow down the streamflow recession ($-dQ/dt$). These competing effects from the

warming climate on $dS/dt$ and $dQ/dS$ would increase the nonlinearity of the recession ($b$) and reduce streamflow stability



[log(*a*)] in cold climate regions such as YRB. In comparison, in the warm climate area, the effect of storage decrease (d*S*/d*t*) on recession (-d*Q*/d*t*) strengthens and the effect of the recession timescale (1/$\tau$ or d*Q*/d*S*) on the recession weakens. As shown in Fig. 8, when temperature is higher (e.g., large positive $\Delta$T), hydrograph recession (negative $\Delta$Q) is faster along with the faster decline of storage ($\Delta$S).

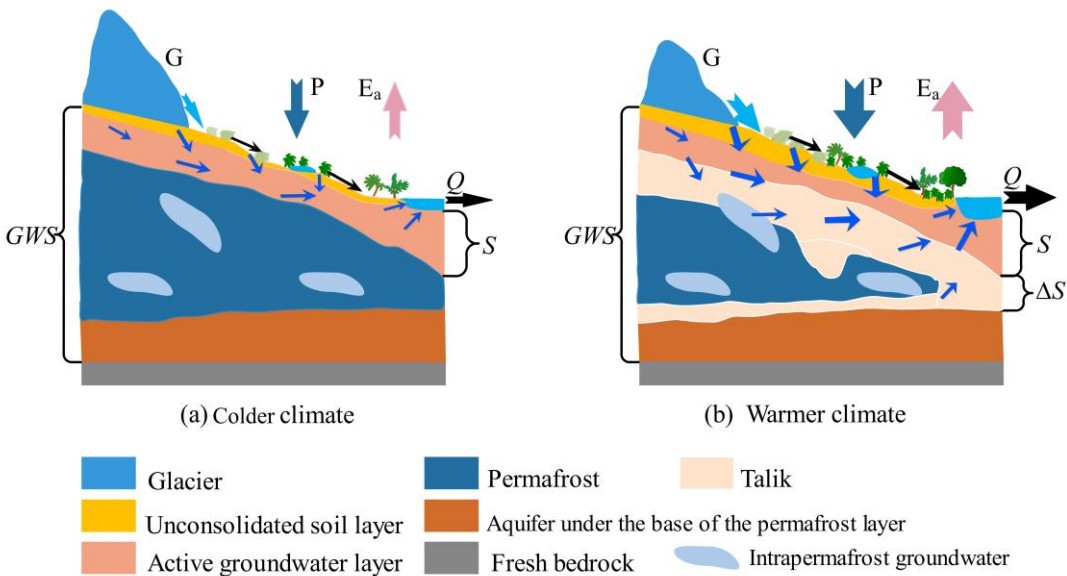


**Figure 9.** A schematic illustration of climate warming effect on surface conditions and subsurface profile as well as hydrological variables. The larger sizes of the arrows indicate a large increase of the hydrological variables, e.g., glacier melting, precipitation, discharge, and evaporation.

Additionally, deep circulating groundwater through macro structures, such as north–south oriented active tensile faults (Fig.
1b) could also affect baseflow and its recharge and discharge (Tan et al., 2021). According to studies using multi-tracer data (e.g., $^2$H, $^3$H, $^{18}$O, and Sr), modern meltwater is found to primarily maintain the rapid recharge of phreatic groundwater in alpine regions through faults and fissures (Shi et al., 2021). In the middle of YRB (i.e., NGS-YC), changes of storage sensitivity to temperature ($\lambda_T$ in Table 4) and recession timescale ($\tau$) are greater than those in the upstream and downstream areas (NGS and NX, respectively). Rising temperature can greatly increase storage (Fig. 8b).

Finally, anthropogenic effect such as reservoir regulation can reduce the climate warming effect on these storage-discharge responses in YRB. For example, in the sub-basin of LS, operations of two reservoirs, PD and ZK, significantly reduced the sensitivity of the recession parameters *a* and *b* to climate warming, and increased streamflow stability [log(*a*)]. It remains questionable however as how this human effort in water management in YRB would be practical/beneficial after the point



when the increase of water storage from glacier and permafrost melt has exhausted the solid volume of water resources in the basin following the climate warming.

## 6. Concluding remarks

Climate warming accelerated after 1997 in YRB of south Tibetan Plateau, especially in its cold and high-altitude upstream areas. Since 1997, the mean annual temperature has risen by 0.75–1.52 °C, and the mean temperature in the annual recession period (1 October – 15 February of the following year) has risen by 1.40–1.78 °C in the five sub-basins of YRB. The larger rise of temperature occurred in the drier and colder sub-basins in the upstream YRB. The recent strong warming has accelerated glacier melting and permafrost thawing, and thereby increased annual streamflow (12.7–31.5% larger than the mean value in the early period before 1997) and streamflow in the recession period (20.9–25.8% larger than before 1997) for the five sub-basins, except LS where reservoir operations are active. These processes initiated by climate warming have changed the hydrological properties of sub-basins considerably and altered the recession characteristics and the storage-discharge relationships.

It has been found that the recession parameter $a$ that characterizes the stability of streamflow has decreased exponentially in the sub-basins, except LS. Meanwhile, the other parameter $b$ that describes the nonlinearity of the recession to discharge increases exponentially in all the sub-basins. These results indicate that climate warming increases the nonlinearity of the recessions and reduces streamflow stability in most of the sub-basins in YRB. Our sensitivity analysis further shows the decrease of the sensitivity of discharge/streamflow to storage under the warming climate. Currently, the accelerated glacier melting and permafrost thawing have recharged the system, thickening the active subsurface zone and increasing groundwater storage. Only a small fraction of the enlarged storage is released in surface streams because the increase of active water layer lengthens subsurface flow paths. These changes have also increased the recession timescale particularly in high altitude cold climate areas. In the warm climate areas downstream of YRB, effect of these changes is minor.

As the liquid water storage has increased greatly from melting glaciers and thawing permafrost in YRB in the recent warming climate, the fast erosion of the solid water storage has weakened its buffering effect of the streamflow which is becoming less stable and more vulnerable to individual intense precipitation events. There are two potential consequences from these changes: one is the increase of flush flooding in the trend of rising precipitation in the high-altitude sub-basins where more land is exposed after the retreat of glaciers, and the other would be the extreme scenario of exhaustion of the water resources in the upstream of YRB after the buffering effect of glacier and permafrost is lost following the continued warming of the climate. While human interference with these processes, via reservoirs and regulations, can reduce and curb these impacts of climate warming on storage-discharge relationships, recession characteristics, and streamflow in short term, as shown in the sub-basin LS, long-term strategies need to be developed to not only cope with the short-term needs but also the sustainability of water resources in the Tibetan Plateau under the threat of the continued warming that could change the entire hydrological system in this critical source region of water for the world most populated nations.



*Author contributions.*

**Jiarong Wang:** Writing-original draft, Investigation, Methodology, Data curation, Visualization. **Xi Chen:** Conceptualization, Writing-review & editing, Formal analysis, Funding acquisition. **Man Gao:** Methodology, Data curation. **Qi Hu:** Writing-review & editing. **Jintao Liu:** Data curation, Validation.

*Competing interest.*

The authors declare that they have no conflict of interest.

*Acknowledgments.*

The research work presented in this paper was supported by the National Natural Science Foundation of China (NSFC) (No. 91747203). Qi Hu's contribution was supported by USDA Cooperative Research Project NEB-38-088.

*Code and data availability.*

All codes and results developed and produced throughout this paper are available upon request to the main author. The sources of data used in this paper were listed in Table 1 and are accessible at the given websites.



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
