# Peer review of "Changes of Nonlinearity and Stability of Streamflow Recession Characteristics under Climate Warming in a Large Glaciated Basin of the Tibetan Plateau"

_Hydrology and Earth System Sciences, 2022_

## Referee Comment (RC2)

**General comments:**

This study mainly analyzed in detail the changes in of nonlinearity and stability of streamflow recession characteristics under climate warming induced by climate variation in the Yarlung-Zangbo River basin (YRB) in the Tibetan Plateau, and the spatial divergency of the impact of climate variation between five sub-basins in YRB. The authors did a very detailed research on streamflow recession characteristics changes in the YRB, and the manuscript was well-written and easy to follow. But there are still some problems to be improved. It is acceptable for publication after minor revisions.

Additional evidence, such as the changes in total days with the mean temperature above 0 °C in a hydrological year (or the recession period), to further testify to the changes in recession characteristics under climate warming. I believe these explanations could strengthen the manuscript quality.

**Minor revisions recommended:**

Line 291. Figure 4: The data points of -dQ/dt~Q are usually scattered to some extent as observation errors and other disturbance in stream and catchment. However, there are pretty concentrated and regular in figure 4. I guess the presented data points of -dQ/dt~Q are more likely extracted from fitted recession segments of Q~t instead of observed hydrograph. The data points of -dQ/dt~Q should be directly calculated from observed hydrograph.

Line 40. It is weird to put the spatial resolution and timescale of data in one column in Table 1. Another column for timescale of data is better.

Variable symbols should keep italic type throughout the manuscript.

The reference part should be further improved according to the demand of the HESS.

---

## Community Comment (CC1)

Dear Editor,

In their interesting paper, "Changes of Nonlinearity and Stability of Streamflow Recession Characteristics under Climate Warming in a Large Glaciated Basin of the Tibetan Plateau", the authors examine changes in the parameters $a$ and $b$ of the power law recession equation given by $-dQ/dt = aQ^b$. Between two periods of years (1980-1996 and 1997-2015), they calculate an increase in $b$ in five basins and a decrease in $a$ in four of five basins. To the overall decrease in $a$, or $\log(a)$, they ascribe a physical significance: a decrease in "streamflow stability".

However, no such physical significance to be ascribed to changes in $a$ alone when $b$ also changes. The problem arises from the units $a$, which change as $b$ changes. The authors are making a nonsensical comparison of two values with different units and claiming one value is less than another.

A consequence of the scale dependence of $a$ is that the reported change in $a$ over time is dependent on the units the authors use for discharge $Q$. If they were to use different units (in other words, rescale $Q$), not only might the absolute and relative magnitudes of the change in $a$ be different, so could the sign of the change (and zero change is also possible given the precise rescaling).

We can take basin YBJ as an example. Table 3 shows values of $a$ decreasing from 0.043 to 0.034 when $Q$ has units of mm/day. At the same time, $b$ increases from 1.79 to 1.90. Converting units of $Q$ to m/day, and keeping $b$ at 1.79 and 1.90, results in values of $a$ of 9.41 and 17.04, respectively (assuming the relationship $-dQ/dt = aQ^b$ holds exactly). Converting units from mm/day to m/day doesn't simply change the values of $a$, but results in an *increase* in $a$ instead of a *decrease* over time. Certainly, if the reported changes in $a$ had a physical significance, simply changing the units wouldn't change the physical interpretation.

The same general problem of misinterpretation exists in their examination of $a$ as a function of temperature.

I recommend that the authors be very careful in their interpretation of changes in $a$ under simultaneous changes in $b$. I also recommend the authors look to Dralle et al. (2015) and Biswal (2021) for further discussion on the relationship between the power law coefficients.

Lastly, on the more general topic of the role of climate on the variability of the $b$ parameter, the authors could look to Jachens et al. (2020) for additional discussion.

Sincerely,
David E. Rupp

**References**

Dralle, D., Karst, N., & Thompson, S. E. (2015). a, b careful: The challenge of scale invariance for comparative analyses in power law models of the streamflow recession. Geophysical Research Letters, 42(21), 9285-9293. https://doi.org/10.1002/2015GL066007

Biswal, B. (2021). Decorrelation is not dissociation: there is no means to entirely decouple the Brutsaert-Nieber parameters in streamflow recession analysis. Advances in Water Resources, 147, 103822. https://doi.org/10.1016/j.advwatres.2020.103822

Jachens, E. R., Rupp, D. E., Roques, C., & Selker, J. S. (2020). Recession analysis revisited: Impacts of climate on parameter estimation. Hydrology and Earth System Sciences, 24(3), 1159-1170. https://doi.org/10.5194/hess-24-1159-2020

---

## Author Comment (AC1)

Dear Editor,

In their interesting paper, "Changes of Nonlinearity and Stability of Streamflow Recession Characteristics under Climate Warming in a Large Glaciated Basin of the Tibetan Plateau", the authors examine changes in the parameters $a$ and $b$ of the power law recession equation given by -$dQ/dt = aQ^b$. Between two periods of years (1980-1996 and 1997-2015), they calculate an increase in $b$ in five basins and a decrease in $a$ in four of five basins. To the overall decrease in $a$, or $\log(a)$, they ascribe a physical significance: a decrease in "streamflow stability".

However, no such physical significance to be ascribed to changes in $a$ alone when $b$ also changes. The problem arises from the units $a$, which change as $b$ changes. The authors are making a nonsensical comparison of two values with different units and claiming one value is less than another.

A consequence of the scale dependence of $a$ is that the reported change in $a$ over time is dependent on the units the authors use for discharge $Q$. If they were to use different units (in other words, rescale $Q$), not only might the absolute and relative magnitudes of the change in $a$ be different, so could the sign of the change (and zero change is also possible given the precise rescaling).

We can take basin YBJ as an example. Table 3 shows values of $a$ decreasing from 0.043 to 0.034 when $Q$ has units of mm/day. At the same time, $b$ increases from 1.79 to 1.90. Converting units of $Q$ to m/day, and keeping $b$ at 1.79 and 1.90, results in values of $a$ of 9.41 and 17.04, respectively (assuming the relationship -$dQ/dt = aQ^b$ holds exactly). Converting units from mm/day to m/day doesn't simply change the values of $a$, but results in an *increase* in $a$ instead of a *decrease* over time. Certainly, if the reported changes in $a$ had a physical significance, simply changing the units wouldn't change the physical interpretation.

The same general problem of misinterpretation exists in their examination of $a$ as a function of temperature.

I recommend that the authors be very careful in their interpretation of changes in $a$ under simultaneous changes in $b$. I also recommend the authors look to Dralle et al. (2015) and Biswal (2021) for further discussion on the relationship between the power law coefficients.

Lastly, on the more general topic of the role of climate on the variability of the $b$ parameter, the authors could look to Jachens et al. (2020) for additional discussion.

Sincerely,
David E. Rupp

**References**

Dralle, D., Karst, N., & Thompson, S. E. (2015). a, b careful: The challenge of scale invariance for comparative analyses in power law models of the streamflow recession. Geophysical Research Letters, 42(21), 9285-9293. https://doi.org/10.1002/2015GL066007.

Biswal, B. (2021). Decorrelation is not dissociation: there is no means to entirely decouple the Brutsaert-Nieber parameters in streamflow recession analysis. Advances in Water Resources, 147, 103822. https://doi.org/10.1016/j.advwatres.2020.103822.

Jachens, E. R., Rupp, D. E., Roques, C., & Selker, J. S. (2020). Recession analysis revisited: Impacts of climate on parameter estimation. Hydrology and Earth System Sciences, 24(3), 1159-1170. https://doi.org/10.5194/hess-24-1159-2020.

**Reply:** We thank Dr. Rupp for his comments and suggestions that help us to reevaluate the recession methodology used in our study.

We agree with his comment that "A consequence of the scale dependence of $a$ is that the reported changes in $a$ over time is dependent on the units for discharge $Q$." The scatterplot of the values of $\log(a)$ and $b$ according to Eq. (1) in our original manuscript is shown in Fig. r1 (a)-(e). It shows strong and significant correlation between $\log(a)$ and $b$ values. Their correlation coefficient $r$ ranges from 0.78 to 0.89 for the five sub-basins in YRB.

We recalculated the parameter $a'$ after scaling $a$ with $q_0$ and $a''$ for fixed $b$ in each sub-basin based on the recession approaches in the two articles recommended by Dr. Rupp (Dralle et al., 2015 and Biswal 2021). We found that the exponential decrease of $a'$ and $a''$ in response to the rise of temperature still exists for the sub-basins except for LS which is affected by reservoir regulations. We interpret our recalculation procedures and results as the follows.

(1) The rescale method

According to Dralle et al. (2015), we rescaled discharge $Q$ by $Q = k\hat{Q}$ and obtained a power law relationship for the rescaled discharge $\hat{Q}$

$$\frac{d\hat{Q}}{dt} = -ak^{b-1}\hat{Q}^b = -a'\hat{Q}^b \tag{1}$$

where $k$ is a constant and $a' = ak^{b-1}$ is a new recession parameter independent of $b$, and the unit of $a'$ is day$^{-1}$ (Dralle et al. 2015).

In order to minimize the correlation of the fitted recession exponents and log-transformed fitted recession scale parameters for a unique value of $q_0$, we use the following equation to compute the scaling factor $q_0$ (Bergner and Zouhar, 2000):

$$q_0 = \exp\left(-\frac{\sum_{i=1}^n (b_i - \bar{b})(\log(a_i) - \overline{\log(a)})}{\sum_{i=1}^n (b_i - \bar{b})^2}\right) \tag{2}$$

where $\bar{b}$ and $\overline{\log(a)}$ are the arithmetic means of annually fitted recession exponents $\{b_1, b_2, \dots, b_n\}$ and log-transformed fitted recession intercepts $\{\log(a_1), \log(a_2), \dots, \log(a_n)\}$, respectively, and $i$ is the number of annual values from 1980 to 2015.

The calculated $q_0$ from (2) is 0.527, 0.602, 0.740, 0.594, and 0.611 mm·day$^{-1}$ for the sub-basins of NGS, YC, NX, YBJ, and LS, respectively. After scaling with $q_0$, the scatterplot of $\log(a')$ and $b$ is shown in Fig. r1 (f)-(j). As expected, there are no correlations between the two recession parameters for the study sub-basins.

[Figure]

Figure r1: (a)-(e) Scatterplots of the values of log($a$) and $b$, and (f)-(j) scatterplots of log($a'$) (after scaling with $q_0$) and $b$ in years from 1980 to 2015 for the sub-basins of NGS, YC, NX, YBJ, and LS.

However, the scatterplot of $a'$ (after scaling with $q_0$) and mean temperature ($T_{re}$) during recession period (Fig. r2) shows that $a'$ still decreases exponentially with rising $T_{re}$ for the sub-basins except for LS with reservoir regulations.  Also, the mean $a'$ values in the recent warmer period of 1997–2015 are smaller than those in the previous period of 1980–1996 for the sub-basins NGS, YC, NX, and YBJ as listed in Table A1.

After transforming into the correlation free representation, the $b$ values do not change as stated by

Dralle et al. (2015).   So, $b$ exponentially increases with the rise of $T_{re}$ for the five sub-basins as shown in our original manuscript.

[Figure]

Figure r2: The exponential function of $a'$ (after scaling with $q_0$) with mean temperature ($T_{re}$) during recession period.   The solid and open circles represent 4-year average and annual value, respectively.

(2) The fixed $b$ method

According to the approach proposed by Biswal (2021), we selected the median of annual $b$ in the period of 1980-2015 as the fixed $b$ value for each sub-basin (Table A1), and then fitted the values of annual $a$ (i.e. $a''$ in Table A1) using the method in our original manuscript (Section 3.2).

The scatterplot of $a''$ (under fixed $b$) and the mean temperature ($T_{re}$) during recession period (Fig. r3) also shows that $a''$ decreases exponentially with the rising $T_{re}$ for the sub-basins except for LS. The mean $a''$ values in the recent warmer period of 1997-2015 are smaller than those in the previous period of 1980-1996 for the sub-basins of NGS, YC, NX, and YBJ as listed in Table A1.

[Figure]

Figure r3: The exponential function of $a''$ (under fixed $b$) with mean temperature ($T_{re}$) during recession period. The solid and open circles represent 4-year average and annual value, respectively.

(3) The physical interpretations

The exponential decrease of annual values of $a$, $a'$ (after scaled with $q_0$), and $a''$ (under fixed $b$), and the increase of $b$ with the rise of $T_{re}$ in the study sub-basins reflect a physical significance of the recession behavior due to climate warming in our study region.

We will revise the manuscript and add detail as follows: the accelerated glacier melting and permafrost thawing have increased the effective hydraulic properties (Lamontagne-Hallé et al., 2018) and the soil active layer thickness (ALT) for groundwater storage. The increase of hydraulic conductivity reduces the buffering effect of soils on streamflow variability and thereby increases the baseflow recession rate. This phenomenon can be identified from the observed hydrographs which suggest that the initial recessions are faster in the warmer period of 1997-2015 for the sub-basins of NGS, YC, NX, and YBJ (Fig. r4). On the other hand, the increase of ALT strengthens aquifer regulations on groundwater flow and slows down the recession rate. This phenomenon can be distinguished in the late slow recession period as shown in Fig. r4. The decrease of $a$ and $a'$ and the increase of $b$ with the rise of temperature suggest a decrease in streamflow stability and an increase of nonlinearity in time in the study region.

As to the report by Jachens et al. (2020), they suggested that the recession parameters assessed by considering the average (or median) values of $a$ and $b$ do not represent watershed properties as much as they represent the climate, and proper evaluation of watershed properties is only ensured by considering independent individual recession events. In our study sub-basins, however, since there is a single hydrograph in a year, the discharge recedes in a long period of time (from September to February of the following year). Thus, the annual variations of the $a$ or $a'$ and $b$ values from 1980 to 2015 in our study sub-basins can represent both the watershed properties and the climate.

Table A1: The median value of $a$ (calculated from the data of the original manuscript), and mean annual parameters of $a'$, $a''$, and $b$ during different sub-periods. Here the parameters $a'$ and $a''$ calculated by methods from Dralle et al. (2015) and Biswal (2021). The asterisk indicates significance with $p < 0.05$. The values in parentheses refer to the range of annual value.

| Index | Period | Sub-basin | | | | |
|---|---|---|---|---|---|---|
| | | NGS | YC | NX | YBJ | LS |
| $a$ | 1980–2015 | 0.042 (0.033~0.060) | 0.032 (0.025~0.043) | 0.022 (0.019~0.027) | 0.038 (0.025~0.052) | 0.024 (0.018~0.032) |
| | 1980–1996 | 0.046 | 0.035 | 0.023 | 0.043 | 0.022 |
| | 1997–2015 | 0.039 | 0.029 | 0.021 | 0.034 | 0.025 |
| | $\Delta a$ | -0.007 | -0.006 | -0.002 | -0.009 | 0.003 |
| $a'$ | 1980–2015 | 0.015 (0.011~0.025) | 0.015 (0.011~0.025) | 0.017 (0.013~0.022) | 0.025 (0.015~0.043) | 0.017 (0.012~0.022) |
| | 1980–1996 | 0.017 | 0.017 | 0.019 | 0.027 | 0.015 |
| | 1997–2015 | 0.014 | 0.014 | 0.015 | 0.023 | 0.017 |
| | $\Delta a'$ | -0.003* | -0.003* | -0.004* | -0.004* | 0.002 |
| $a''$ | 1980–2015 | 0.056 (0.042~0.094) | 0.039 (0.028~0.063) | 0.026 (0.020~0.034) | 0.047 (0.028~0.089) | 0.023 (0.015~0.032) |
| | 1980–1996 | 0.062 | 0.044 | 0.028 | 0.052 | 0.022 |
| | 1997–2015 | 0.051 | 0.035 | 0.024 | 0.043 | 0.024 |
| | $\Delta a''$ | -0.011* | -0.008* | -0.004* | -0.009* | 0.002 |
| $b$ | 1980–2015 | 1.85 (1.645~1.990) | 1.70 (1.506~1.992) | 1.54 (1.297~1.789) | 1.85 (1.607~1.979) | 1.36 (1.117~1.783) |
| | 1980–1996 | 1.81 | 1.67 | 1.48 | 1.78 | 1.25 |
| | 1997–2015 | 1.89 | 1.73 | 1.59 | 1.90 | 1.37 |
| | $\Delta b$ | 0.08* | 0.06 | 0.11* | 0.11* | 0.12* |
| | Median | 1.900 | 1.820 | 1.591 | 1.894 | 1.329 |

[Figure]

Figure r4: The discharge recession for the selected years with approximately the same initial discharge $Q_0$ in each sub-basin.

**References**

Bergner, F., and G. Zouhar.: A new approach to the correlation between the coefficient and the exponent in the power law equation of fatigue crack growth, Int. J. Fatigue, 22(3), 229–230, https://doi:10.1016/S0142-1123(99)00123-1, 2000.

Biswal, B.: Decorrelation is not dissociation: there is no means to entirely decouple the Brutsaert-Nieber parameters in streamflow recession analysis. Advances in Water Resources, 147, 103822, https://doi.org/10.1016/j.advwatres.2020.103822, 2021.

Dralle, D., Karst, N., & Thompson, S. E.: a, b careful: The challenge of scale invariance for comparative analyses in power law models of the streamflow recession. Geophysical Research Letters, 42(21), 9285-9293, https://doi.org/10.1002/2015GL066007, 2015.

Jachens, E. R., Rupp, D. E., Roques, C., & Selker, J. S.: Recession analysis revisited: Impacts of climate on parameter estimation. Hydrology and Earth System Sciences, 24(3), 1159-1170, https://doi.org/10.5194/hess-24-1159-2020, 2020.

Lamontagne-Hallé, P., McKenzie, J. M., Kurylyk, B. L., and Zipper, S. C.: Changing groundwater discharge dynamics in permafrost regions, Environ. Res. Lett., 13, 084017, https://doi.org/10.1088/1748-9326/aad404, 2018.

---

## Author Comment (AC3)

**Comment on hess-2022-25**
**Anonymous Referee #2**

**General comments:**

This study mainly analyzed in detail the changes in of nonlinearity and stability of streamflow recession characteristics under climate warming induced by climate variation in the Yarlung-Zangbo River basin (YRB) in the Tibetan Plateau, and the spatial divergency of the impact of climate variation between five sub-basins in YRB.

The authors did a very detailed research on streamflow recession characteristics changes in the YRB, and the manuscript was well-written and easy to follow. But there are still some problems to be improved. It is acceptable for publication after minor revisions.

Additional evidence, such as the changes in total days with the mean temperature above 0 °C in a hydrological year (or the recession period), to further testify to the changes in recession characteristics under climate warming. I believe these explanations could strengthen the manuscript quality.

**Reply:** We calculated the total number of days with mean temperature above 0°C (MTD) in a year and the recession period, respectively, for the five sub-basins (Fig. R1). The annual MTD increases significantly at a rate of 0.48~0.82 days·a$^{-1}$ in the sub-basins. The total of mean MTD in the recent period of 1997 ~ 2015 is 8~18 days greater than that in the early period of 1980 ~ 1996 (Fig. R1a). Meanwhile, the annual MTD in the recession period increases, tested to be significant in the mainstream of YRB (e.g., NGS, YC and NX), and insignificant in the two sub-basins of YBJ and LS. The multiyear mean MTD in the recession period is 2~7 days greater in the recent period than in the early period (Fig. R1b).

[Figure]

Figure R1: The total number of days with the mean temperature above 0 °C (MTD) in a year (a) and the recession period (b) from 1980 to 2015 in the five sub-basins. The subscripts "1" and "2" refer to the early period from 1980 to 1996 and the recent period from 1997 to 2015, respectively. * is significant tested by TFPW-MK ($p < 0.05$).

The increased MTD promotes thawing of the frozen ground, and thereby increases the active soil layer thickness (ALT, as shown in Fig. 2h in the original manuscript). Eventually, climate warming decreases streamflow stability and increases nonlinearity of the hydrographs in the study sub-basins.

**Minor** comments

1. Line 291. Figure 4: The data points of -d$Q$/d$t$~$Q$ are usually scattered to some extent as observation errors and other disturbance in stream and catchment. However, there are pretty concentrated and regular in figure 4. I guess the presented data points of -d$Q$/d$t$~$Q$ are more likely extracted from fitted recession segments of $Q$~$t$ instead of observed hydrograph. The data points of -d$Q$/d$t$~$Q$ should be directly calculated from observed hydrograph.

**Reply:** For Figs. 4a-4e in the original manuscript, the data points of -d$Q$/d$t$~$Q$ are extracted from the fitted recession segments of $Q$~$t$. We have redrawn the figures of -d$Q$/d$t$~$Q$ using the observed hydrographs and then fitted the lines in each of the two periods (1980-1996 and 1997-2015) (see Fig. R2). These fitted values of the recession parameters ($b$ and $\log(a)$) are different with those of Figs. 4a-4e in the original manuscript, but changes of $b$ and $\log(a)$ between the two periods are consistent with those in Figs. 4a-4e. So, it does not affect our conclusions of changes in streamflow recession characteristics under climate change.

We will adapt Fig R2 in our revised manuscript.

[Figure]

Figure R2: (a)-(e): Plot of -d$Q$/d$t$ vs. $Q$ in log-log space for recession hydrographs during 1980–2015, and the fitting lines [$\log(-dQ/dt) = b\log(Q)+\log(a)$] for the data points in the two periods (1980-1996 and 1997-2015) for the five sub-basins. (f): Differences of mean recession rates between the two periods ($\Delta v_Q$) estimated from the non-overlapping moving averages of the 5-days' series.

2. Line 40. It is weird to put the spatial resolution and timescale of data in one column in Table 1. Another column for timescale of data is better.

**Reply:** Table 1 is revised as the follows.

Table 1. Information of the data used in this study.

| Data | Period | Spatial-Resolution | Temporal-Resolution | Source |
|---|---|---|---|---|
| Precipitation (P, mm) Mean Temperature (T, ℃) Evapotranspiration (E, mm) Discharge (Q, mm) | 1980~2015 | 0.1°×0.1° Obs. stations | Daily Daily | National Tibetan Plateau Data Center; http://data.tpdc.ac.cn http://data.cma.cn |
| NDVI | 1982-2015 | 1/12°×1/12° | 15-days | http://data.tpdc.ac.cn |
| Glacial area | 1976, 2000, 2013 | 30m×30m | Annual | http://data.tpdc.ac.cn and China's second glacier catalogue data |
| | 2006~2011 (in 2009) | | Mean annual | |
| Permafrost and Frozen ground | 1983-1996, 1997, 2003, 2012, 2017 | 1km×1km | Mean annual | http://data.tpdc.ac.cn |
| Active layer thickness (ALT) | 1980-2015 | 0.1°×0.1° | Annual | Calculated by a linear function from Xu et al. (2017) |

3. Variable symbols should keep italic type throughout the manuscript.

**Reply:** We will revise the relevant variables using an italic type in the manuscript.

4. The reference part should be further improved according to the demand of the HESS.

**Reply:** We will revise the references in terms of the HESS formations.

---

## Author Response (AR1)

Dear Editor,

**RE:** Manuscript #hess-2022-25 "Changes of Nonlinearity and Stability of Streamflow Recession Characteristics under Climate Warming in a Large Glaciated Basin of the Tibetan Plateau".

We thank two anonymous reviewers and Dr. Rupp for their valuable comments that have helped us to improve the manuscript. The revised sentences and sections in the revised manuscript are highlighted in blue color. The major modifications are summarized as follows.

(1) According to the comments of Dr. Rupp, we recalculated the parameter $a'$ after scaling $a$ with $k$ in each sub-basin using the decorrelation method proposed by (Dralle et al., 2015) and rederived the corresponding equations (Lines 189-197 and Tables 3 and 4 in our revised manuscript). As shown in the replies to Dr. Rupp's comments, the exponential decrease of $a'$ in response to the rise of temperature still exists for the sub-basins except LS.

(2) According to the comments of Reviewer 1, we expanded our discussion of the physical meanings and the driving forces of changes for the recession parameters of $a$ ($a'$) and $b$ from the aspects of increased soil active layer thickness and climate warming.

(3) According to the comments of Reviewer 2, we added the total number of days with mean temperature above 0°C in a year ($MTD_a$) and the recession period ($MTD_{re}$) revised Figs. 2j and 2k, respectively.

Other minor issues raised by the reviewers have all been addressed accordingly.

We hope that the revision has addressed all the concerns of the reviewers.

Thank you for your editorial work.

Sincerely,

Xi Chen
On behalf of all co-authors

Xi Chen, Professor of Hydrology
Institute of Surface-Earth System Science,
Tianjin University, Tianjin 300072, China
E-mail: xichen@hhu.edu.cn

**Comment on hess-2022-25**

**Anonymous Referee #1**

Referee comment on "Changes of Nonlinearity and Stability of Streamflow Recession Characteristics under Climate Warming in a Large Glaciated Basin of the Tibetan Plateau" by Jiarong Wang et al., Hydrol. Earth Syst. Sci. Discuss., https://doi.org/10.5194/hess-2022-25-RC1, 2022.

In cold alpine regions, climate warming has changed infiltration and hydraulic connectivity due to accelerated glacier melting and permafrost thawing as well as significant glacier and permafrost retreats. It should later the hydrograph pattern including the recession process. Authors analyzed the temporal changes of the recession parameters of a and b in the Brutsaert and Nieber equation in terms of the daily observed discharge during 1980–2015 in the Yarlung-Zangpo River basin (YRB). They obtained interesting results that $a$ decreased and $b$ increased with air temperature rise, meaning increase of nonlinearity and decrease of stability for the streamflow recessions in most sub-basins of YRB due to climate warming. This finding will benefit to establish a method for hydrological prediction and baseflow analysis in cold watersheds.

The manuscript was well-written and easy to follow. It is acceptable for publication after minor revisions.

Since changes of $a$ and $b$ values are highly related to the enlarged groundwater storage or soil active layer thickness, I suggested that authors to clearly state the physical bases of the changes of a and b. I also suggested to explain how the driving forces or changes of soil active layer thickness lead to the initially fast decline of recession (ascribed to the increased $b$) and finally slow decline of recession (ascribed to the decreased $a$). I believe these explanations could strengthen the manuscript quality.

**Reply:** We thank this anonymous reviewer for the valuable comments that helped us to improve our manuscript.

As shown in the observed hydrographs in Fig. R1 the streamflow recedes fast in the early phase of the recession and slows down in the later phase. In addition, Fig. R1 shows that the recession rate ($-dQ/dt$) is small (large) for small (large) streamflow in our study sub-basins.

We have revised the manuscript and add detail as follows: the accelerated glacier melting and permafrost thawing have increased the effective hydraulic properties (Lamontagne-Hallé et al., 2018) and the soil active layer thickness (ALT) for groundwater storage. The increase of hydraulic conductivities reduces the buffering effect of soils on streamflow variability and thereby increases the baseflow recession rate. This phenomenon can be identified in the observed hydrographs which show that the streamflow in the early phase of recession is faster in the warmer period of 1997-2015 in sub-basins NGS, YC, NX, and YBJ (Fig. R1). The warming-resulted increase of ALT strengthens aquifer regulations on groundwater flow so to slow down the recession rate as the warm season proceeds. This weakening of streamflow in the late phase of the recession is also shown in Fig. R1. So, the decrease of $a$ (and $a'$ which is a new recession parameter independent of $b$ in Eq. (8) in revised manuscript, line 191) and the increase of $b$ with the rise of temperature can illustrate an increase in streamflow stability and nonlinearity in time in the study basins.

[Figure]

Figure R1: The discharge recession for the selected years with approximately the same initial discharge $Q_0$ in the study sub-basins.

**Minors**

1. The decrease of log($a$) means the decrease of recession rate and thus increase of the streamflow stability, right?
**Reply:** Defined by Tashie et al. (2019), "an increase in the value of $a$ increases rates of streamflow "decay", while the value of $b$ is a measure of "nonlinearity" with greater nonlinearity enhancing the concavity of the hydrograph." A larger $a$ value indicates a greater recession rate in the log ($a$)-$t$ relationship (see Fig. 3 in Tashie et al. 2019). So, the decrease of log($a$) means the decrease of recession rate and the increase of streamflow stability.

2. Line 100. "…, mean annual temperature varies from -9.3 to 22.0 °C". Is it right the annual temperature could as high as 22.0 °C in the basin?
**Reply:** This is a spatial range of the mean annual temperature across the Yarlung-Zangpo River basin (YRB) from the west to the east. The mean annual temperature in the downstream valleys could be as high as 22°C.

3. Line 104. "Groundwater accounts for about 54% of the annual streamflow". References are needed.
**Reply:** The reference has been added. We have revised as "Groundwater accounts for about 55% of the annual streamflow in upstream and 27% in downstream of YRB (Yao et al., 2021)" from lines 111 to 112 in revision.

4. Fig 3. There is a mistake for the range of mean daily precipitation
**Reply:** The range of daily precipitation in Fig. 3 comes from the observed daily precipitation data in the two periods 1980-1996 and 1997-2015. There is a mistake that the lower bound of the daily precipitation was not shown in that figure. We have redrawn the figure shown below and included

the revised figure in our revision.

[Figure]

Figure 3: (a)-(e): Mean daily precipitation *P*, temperature *T*, and discharge *Q* in a hydrological year (from 1 March to 28 February of the following year) for the two periods in the five sub-basins. The red dashed rectangle in (a) shows the hydrograph recession from 1 October to 15 February of the following year, and the shading shows the range of the daily variation of *P*, *T,* and *Q* in each period.

**References:**

Lamontagne-Hallé, P., McKenzie, J. M., Kurylyk, B. L., and Zipper, S. C.: Changing groundwater discharge dynamics in permafrost regions, Environ. Res. Lett., 13, 084017, https://doi.org/10.1088/1748-9326/aad404, 2018.

Tashie, A. M., Scaife, C. I., & Band, L. E.: Transpiration and subsurface controls on streamflow recession characteristics. Hydrological Processes, 33(19), 2561–2575. https://doi.org/10.1002/hyp.13530, 2019.

Yao, Y., Zheng, C., Andrews, C. B., et al.: Role of groundwater in sustaining northern Himalayan Rivers. Geophysical Research Letters, 48, e2020GL092354. https://doi.org/10.1029/2020GL092354, 2021.

**Comment on hess-2022-25**
**Anonymous Referee #2**

**General comments:**
This study mainly analyzed in detail the changes in of nonlinearity and stability of streamflow recession characteristics under climate warming induced by climate variation in the Yarlung-Zangbo River basin (YRB) in the Tibetan Plateau, and the spatial divergency of the impact of climate variation between five sub-basins in YRB.

The authors did a very detailed research on streamflow recession characteristics changes in the YRB, and the manuscript was well-written and easy to follow. But there are still some problems to be improved. It is acceptable for publication after minor revisions.

Additional evidence, such as the changes in total days with the mean temperature above 0 ℃ in a hydrological year (or the recession period), to further testify to the changes in recession characteristics under climate warming. I believe these explanations could strengthen the manuscript quality.

**Reply:** We calculated the total number of days with mean temperature above 0°C in a year (MTD$_a$) and the recession period (MTD$_{re}$), respectively, for the five sub-basins (Fig. R2). The annual MTD$_a$ increases significantly at a rate of 0.48~0.82 days·a$^{-1}$ in the sub-basins. The total of mean MTD$_a$ in the recent period of 1997 ~ 2015 is 8~18 days greater than that in the early period of 1980 ~ 1996 (Fig. R2a). Meanwhile, the annual MTD$_{re}$ in the recession period increases, tested to be significant in the mainstream of YRB (e.g., NGS, YC and NX), and insignificant in the two sub-basins of YBJ and LS. The multiyear mean MTD$_{re}$ in the recession period is 2~7 days greater in the recent period than in the early period (Fig. R2b). MTD$_{re}$ increases significantly at a rate of 0.28~0.32 days·a$^{-1}$ in the mainstream of YRB, and increases insignificantly at a rate of 0.12, 0.17 days·a$^{-1}$ in YBJ and LS, respectively. We have added MTD$_a$ and MTD$_{re}$ in Figs. 2j and 2k in our revision, respectively.

[Figure]

Figure R2: Variations of the total number of days with the mean temperature above 0 ℃ in a year (MTD$_a$, a) and the recession period (MTD$_{re}$, b) from 1980 to 2015 in the five sub-basins. The subscripts "1" and "2" refer to the early period from 1980 to 1996 and the recent period from 1997 to 2015, respectively. **\*** is significant tested by TFPW-MK ($p < 0.05$).

The increased MTD promotes thawing of the frozen ground, and thereby increases the active soil layer thickness (ALT, as shown in Fig. 2h in the original manuscript). Eventually, climate warming decreases streamflow stability and increases nonlinearity of the hydrographs in the study sub-basins.

**Minor** comments

1. Line 291. Figure 4: The data points of -d$Q$/d$t$~$Q$ are usually scattered to some extent as observation errors and other disturbance in stream and catchment. However, there are pretty concentrated and regular in figure 4. I guess the presented data points of -d$Q$/d$t$~$Q$ are more likely extracted from fitted recession segments of $Q$~$t$ instead of observed hydrograph. The data points of -d$Q$/d$t$~$Q$ should be directly calculated from observed hydrograph.

**Reply:** For Figs. 4a-4e in the original manuscript, the data points of -d$Q$/d$t$~$Q$ are extracted from the fitted recession segments of $Q$~$t$. We have redrawn the figures of -d$Q$/d$t$~$Q$ using the observed hydrographs and then fitted the lines in each of the two periods (1980-1996 and 1997-2015) (see Fig. R3). These fitted values of the recession parameters ($b$ and log($a$)) are different with those of Figs. 4a-4e in the original manuscript, but changes of $b$ and log($a$) between the two periods are consistent with those in Figs. 4a-4e. So, it does not affect our conclusions of changes in streamflow recession characteristics under climate change.

We have adapted Fig R3 in our revised manuscript (i.e. Fig. 5, line 320).

[Figure]

Figure R3: (a)-(e): Plot of -d$Q$/d$t$ vs. $Q$ in log-log space for recession hydrographs during 1980–2015, and the fitting lines [log(-d$Q$/d$t$) = $b$log($Q$)+log($a$)] for the data points in the two periods (1980-1996 and 1997-2015) for the five sub-basins. (f): Differences of mean recession rates between the two periods ($\Delta v_Q$) estimated from the non-overlapping moving averages of the 5-days' series.

2. Line 40. It is weird to put the spatial resolution and timescale of data in one column in Table 1. Another column for timescale of data is better.

**Reply:** Table 1 is revised as the follows (line 148).

Table 1. Information of the data used in this study.

| Data | Period | Spatial-Resolution | Temporal-Resolution | Source |
|---|---|---|---|---|
| Precipitation (P, mm)
Mean Temperature (T, ℃)
Evapotranspiration (E, mm)
Discharge (Q, mm) | 1980~2015 | 0.1°×0.1°

Obs. stations | Daily

Daily | National Tibetan Plateau Data Center;
http://data.tpdc.ac.cn
http://data.cma.cn |
| NDVI | 1982-2015 | 1/12°×1/12° | 15-days | http://data.tpdc.ac.cn |
| Glacial area | 1976, 2000, 2013 | 30m×30m | Annual | http://data.tpdc.ac.cn and China's second glacier catalogue data |
| | 2006~2011 (in 2009) | | Mean annual | |
| Permafrost and Frozen ground | 1983-1996, 1997, 2003, 2012, 2017 | 1km×1km | Mean annual | http://data.tpdc.ac.cn |
| Active layer thickness (ALT) | 1980-2015 | 0.1°×0.1° | Annual | Calculated by a linear function from Xu et al. (2017) |

3. Variable symbols should keep italic type throughout the manuscript.

**Reply:** We have revised the relevant variables using an italic type in the manuscript.

4. The reference part should be further improved according to the demand of the HESS.

**Reply:** We have revised the references in terms of the HESS formations.

---

## Editor Decision (ED1)

Hydrol. Earth Syst. Sci. Discuss., referee comment RC1
https://doi.org/10.5194/hess-2022-25-RC1, 2022
**Comment on hess-2022-25**

Anonymous Referee #1
* * *
Referee comment on "Changes of Nonlinearity and Stability of Streamflow Recession Characteristics under Climate Warming in a Large Glaciated Basin of the Tibetan Plateau" by Jiarong Wang et al., Hydrol. Earth Syst. Sci. Discuss., https://doi.org/10.5194/hess-2022-25-RC1, 2022
* * *
In cold alpine regions, climate warming has changed infiltration and hydraulic connectivity due to accelerated glacier melting and permafrost thawing as well as significant glacier and permafrost retreats. It should later the hydrograph pattern including the recession process. Authors analyzed the temporal changes of the recession parameters of a and b in the Brutsaert and Nieber equation in terms of the daily observed discharge during 1980–2015 in the Yarlung-Zangpo River basin (YRB). They obtained interesting results that a decreased and b increased with air temperature rise, meaning increase of nonlinearity and decrease of stability for the streamflow recessions in most sub-basins of YRB due to climate warming. This finding will benefit to establish a method for hydrological prediction and baseflow analysis in cold watersheds.

The manuscript was well-written and easy to follow. It is acceptable for publication after minor revisions.

Since changes of a and b values are highly related to the enlarged groundwater storage or soil active layer thickness, I suggested that authors to clearly state the physical bases of the changes of a and b. I also suggested to explain how the driving forces or changes of soil active layer thickness lead to the initially fast decline of recession (ascribed to the increased b) and finally slow decline of recession (ascribed to the decreased a). I believe these explanations could strengthen the manuscript quality.

Minors

- the decrease of log(a) means the decrease of recession rate and thus increase of the streamflow stability, right?
- Line 100. "…, mean annual temperature varies from -9.3 to 22.0 â□□". Is it right the annual temperature could as high as 22.0 â□□ in the basin?
- Line 104. "Groundwater accounts for about 54% of the annual streamflow". References are needed.
- Fig 3. There is a mistake for the range of mean daily precipitation.

**General comments:**

This study mainly analyzed in detail the changes in of nonlinearity and stability of streamflow recession characteristics under climate warming induced by climate variation in the Yarlung-Zangbo River basin (YRB) in the Tibetan Plateau, and the spatial divergency of the impact of climate variation between five sub-basins in YRB. The authors did a very detailed research on streamflow recession characteristics changes in the YRB, and the manuscript was well-written and easy to follow. But there are still some problems to be improved. It is acceptable for publication after minor revisions.

Additional evidence, such as the changes in total days with the mean temperature above 0 °C in a hydrological year (or the recession period), to further testify to the changes in recession characteristics under climate warming. I believe these explanations could strengthen the manuscript quality.

**Minor revisions recommended:**

Line 291. Figure 4: The data points of -dQ/dt~Q are usually scattered to some extent as observation errors and other disturbance in stream and catchment. However, there are pretty concentrated and regular in figure 4. I guess the presented data points of -dQ/dt~Q are more likely extracted from fitted recession segments of Q~t instead of observed hydrograph. The data points of -dQ/dt~Q should be directly calculated from observed hydrograph.

Line 40. It is weird to put the spatial resolution and timescale of data in one column in Table 1. Another column for timescale of data is better.

Variable symbols should keep italic type throughout the manuscript.

The reference part should be further improved according to the demand of the HESS.